# Distinct functions of microtubules and actin filaments in the transportation of the male germ unit in pollen

Xiangfei Wang [1,3], Tonghui Li[1,3], Jiuting Xu[1,3], Fanfan Zhang[1], Lifang Liu[1], Ting Wang[1], Chun Wang [1], Haiyun Ren [1,2] ✉ & Yi Zhang [1] ✉

Flowering plants rely on the polarized growth of pollen tubes to deliver sperm cells (SCs) to the embryo sac for double fertilization. In pollen, the vegetative nucleus (VN) and two SCs form the male germ unit (MGU). However, the mechanism underlying directional transportation of MGU is not well understood. In this study, we provide the first full picture of the dynamic interplay among microtubules, actin filaments, and MGU during pollen germination and tube growth. Depolymerization of microtubules and inhibition of kinesin activity result in an increased velocity and magnified amplitude of VN's forward and backward movement. Pharmacological washout experiments further suggest that microtubules participate in coordinating the directional movement of MGU. In contrast, suppression of the actomyosin system leads to a reduced velocity of VN mobility but without a moving pattern change. Moreover, detailed observation shows that the direction and velocity of VN's movement are in close correlations with those of the actomyosin-driven cytoplasmic streaming surrounding VN. Therefore, we propose that while actomyosin-based cytoplasmic streaming influences on the oscillational movement of MGU, microtubules and kinesins avoid MGU drifting with the cytoplasmic streaming and act as the major regulator for fine-tuning the proper positioning and directional migration of MGU in pollen.

Double fertilization is a unique and feature-defining process of flowering plants, which represent the most widespread and abundant group of plants on earth today. During double fertilization, each of the two sperm cells (SCs) fuses with one of the two female reproductive cells, the egg cell and central cell, to develop an embryo and endosperm in a developing seed[1]. As SCs of flowering plants have lost motility, their transport to female gametes relies on the directional growth of pollen tubes towards the embryo sac[2,3]. In a typical pollen grain, a vegetative cell encapsulates two SCs. The two SCs remain stably linked until before fertilization, and one of the SCs is consistently associated with the vegetative cell nucleus (VN) through a cytoplasmic projection. These physical interactions unite the SCs and VN to form a structure termed the male germ unit (MGU)[4].

In most flowering plants, including the model plant *Arabidopsis thaliana*, all three members of the MGU travel together during pollen tube growth, with the VN preceding the SCs at a position close to the growing tip of pollen tubes[4,5]. Pollen tubes from *Arabidopsis drop1⁻ drop2⁻* and *cdka;1⁺/⁻ fbl17⁺/⁻* mutants that lack SCs can still grow normally towards the ovule, supporting the common idea that SCs are passive cargos of the pollen tubes[6,7]. These studies also suggested that VN may play a major role in mediating the directional movement of MGU in pollen tube[6,7]. In contrast, when the mobility of VN was

[1]Key Laboratory of Cell Proliferation and Regulation Biology of Ministry of Education, College of Life Sciences, Beijing Normal University, 100875 Beijing, China. [2]Center for Biological Science and Technology, Guangdong Zhuhai-Macao Joint Biotech Laboratory, Beijing Normal University, 519087 Zhuhai, China. [3]These authors contributed equally: Xiangfei Wang, Tonghui Li, Jiuting Xu. ✉e-mail: hren@bnu.edu.cn; yi.zhang@bnu.edu.cn

impaired in an *Arabidopsis* double mutant of the klarsicht/ANC-1/Syne homology (KASH) protein-coding genes, *WIT1* and *WIT2*, SCs led VN as they moved or just moved forward without VN in growing pollen tubes[5,8]. Based on these observations, it was proposed that there might be two driving forces that regulate the movements of the VN and SCs, respectively, and coordinate with each other to move the MGU forward[5]. To date, however, the driving forces and underlying mechanisms of MGU migration during pollen tube growth are still largely unknown.

In mammals, sperm motility relies on microtubules, an integral part of the sperm flagellum[9]. Several pharmacological experiments revealed that MGU migration in plants was also regulated by microtubules. For example, inhibition of microtubule polymerization caused a significant retardation of the movement of MGU from the grain to the tube, an increased distance between VN and generative cells, and an enhanced proportion of pollen tubes with generative cells preceding the VN in *Galanthus nivalis* and tobacco pollen tubes[10–12]. These studies indicated that microtubules were required for the process of MGU entering the pollen tube and maintaining the physical connection between VN and generative cells in MGU. Consistent with these pharmacological studies, immunofluorescence assays and transient transformation of GFP-tagged AtEB1c, a microtubule plus end-binding protein, detected tubulin-accumulated structures around the generative cell or extensive microtubule bundles in tobacco pollen tubes[10,12–14]. However, a stably transformed fluorescent marker line for microtubules is not available, which restricts the study of microtubule dynamics in pollen and the molecular mechanisms of MGU migration mediated by the microtubule cytoskeleton.

Compared with the work on microtubules, extensive studies have been carried out to characterize the roles of actin filaments in pollen. It has been well recognized that the actin cytoskeleton is responsible for generating the driving force of cytoplasmic streaming required for the long-distance transport of organelles and secretory vesicles during polarized growth of pollen tubes[2,15–18]. Moreover, actin polymerization was recently found to be essential for vesicle mobility and polarity establishment in pollen grains[19,20]. However, whether the actin network is involved in the movement of MGUs has not been investigated. An immunofluorescence assay using heterogeneous anti-myosin antibodies revealed that myosin I was localized to the surface of generative cells and VN in lily and tobacco pollen tubes[21], indicating the possible participation of the actin cytoskeleton in MGU migration. Moreover, a recent study observed saltatory SC movement in *Arabidopsis* pollen tubes[22]. The authors assumed that SC movement was mediated by kinesins with a calponin homology domain (KCH) that can bind both actin filaments and microtubules[22]. Collectively, these studies implied that microtubules and actin filaments may both be involved in MGU migration, but their specific functions and how they coordinate have not been studied thus far.

Here, through intensive screening of the localization of tubulins and microtubule-associated proteins, we found that the fluorescently tagged microtubule-binding domain (MBD) of *Arabidopsis* MAP4 could be used to visualize microtubules in pollen grains and tubes. Together with the available marker lines for actin filaments and MGU, we studied the dynamics of microtubules and actin filaments during the movement of MGU. We revealed that the actomyosin-dependent cytoplasmic streaming influences the local movement of MGU, and the microtubule system controls the appropriate positioning and directional movement of MGU in pollen, which guarantees the successful delivery of SCs for double fertilization.

## Results
### Generation of fluorescently labeled reporter lines for visualizing microtubules in pollen
Despite the abundance of microtubules in pollen grains and tubes[23,24], detailed observation of microtubule dynamics during pollen germination and tube elongation has been limited due to the lack of reporters. To generate suitable microtubule reporters in pollen, we first analyzed the subcellular localization of several tubulins with high expression in pollen, including TUA1, TUB1, TUB4, and TUB9 (Supplementary Fig. 1). The C- or N-terminal fluorescently tagged constructs under the control of their native promoters were generated and transformed into the relevant mutant background. The subcellular localization of the expressed fusion proteins is summarized in Supplementary Table 1. In brief, TUA1-GFP, TUB4-GFP, TUB9-GFP and mScarlet-TUB1 were localized in the cytoplasm with some accumulation of signals around the DAPI-stained VN and SCs in pollen grains and tubes (Fig. 1a, c, d, f); TUB1-GFP and mScarlet-TUA1 only displayed weak fluorescence in the cytoplasm (Fig. 1b, e); mScarlet-TUB4 and mScarlet-TUB9 labeled microtubules in pollen tubes, but showed a diffuse cytoplasmic distribution in pollen grains (Fig. 1g, h). We also generated stable transgenic plants that express fluorescently tagged microtubule-associated proteins or domains under the control of a pollen-specific promoter, Lat52[25], to visualize microtubules in pollen (Supplementary Table 1). We observed that mCherry-AtEB1a displayed diffuse localization in the cytoplasm (Fig. 1i), mCherry-AtEB1b localized in the cytoplasm and around the VN (Fig. 1j), and mCherry-AtEB1c was specifically located in the VN (Fig. 1k). We found that mCherry-MBD successfully labeled microtubules in both pollen grains and pollen tubes (Fig. 1l). To confirm that mCherry-MBD is a useful marker for visualizing microtubules in pollen, we performed an immunofluorescence assay using an antibody against β-tubulin and found that the microtubule structures in pollen grains and tubes were roughly similar to those labeled with mCherry-MBD (Fig. 1m). We therefore selected a homozygous line of mCherry-MBD with moderate fluorescence to visualize the organization and dynamics of microtubules in pollen. This and other marker lines, including plants expressing mEGFP-MBD and the markers for the actin cytoskeleton, Lifeact-mCherry, and Lifeact-mEGFP[19], showed similar pollen germination rate, pollen tube growth rate and fertility as the control (Supplementary Fig. 2).

### The dynamics of microtubules and actin filaments in pollen
To simultaneously detect the dynamics of microtubules and actin filaments in pollen, we crossed the mCherry-MBD line with the Lifeact-mEGFP line. Both fluorescent signals were diffused in the cytoplasm before the hydration of pollen grains (Fig. 2a, 0 s). Shortly after hydration, the two cytoskeleton systems initiated their assembly at different sites: microtubules rapidly assembled around sphere-shaped structures in pollen grains (Fig. 2a, 45–135 s, Fig. 2b), whereas the strongest fluorescence of actin filaments first appeared near the plasma membrane (Fig. 2a, 45 s, Fig. 2b) and then presented in the center of pollen grains (Fig. 2a, 135 s, Fig. 2b). The evident filamentous structures labeled by Lifeact-mEGFP at cell cortex were depolymerized by the application of Latrunculin B (LatB), an inhibitor of actin polymerization (Fig. 2c). These observations supported that the Lifeact-mEGFP signal appeared at the plasma membrane was truly derived from actin filaments. Before germination, actin filaments displayed rotational movement mainly on the edge of pollen grains[19] (Fig. 2d, 0 s, Supplementary Movie 1). Nevertheless, microtubules formed a cage-like meshwork, located in the relatively central region and did not rotate with actin filaments (Fig. 2d, 0 s, Fig. 2e, Supplementary Movie 1). When actin filaments developed the collar-like construction at the future germination site, the cage-like network of microtubules contracted behind the actin structure, opposite to the future site of pollen tube outgrowth (Fig. 2d, 960 s, Supplementary Movie 1). After the emergence of pollen tubes, actin filaments entered the tube first, while microtubules followed until the tube grew to a length of $6.5 \pm 1.4\,\mu m$ (mean ± SD, $n = 21$) (Fig. 2d, 2160–2340 s, Fig. 2f, Supplementary Movie 1). In the fast-growing pollen tube, long and thick microtubule bundles extended from the

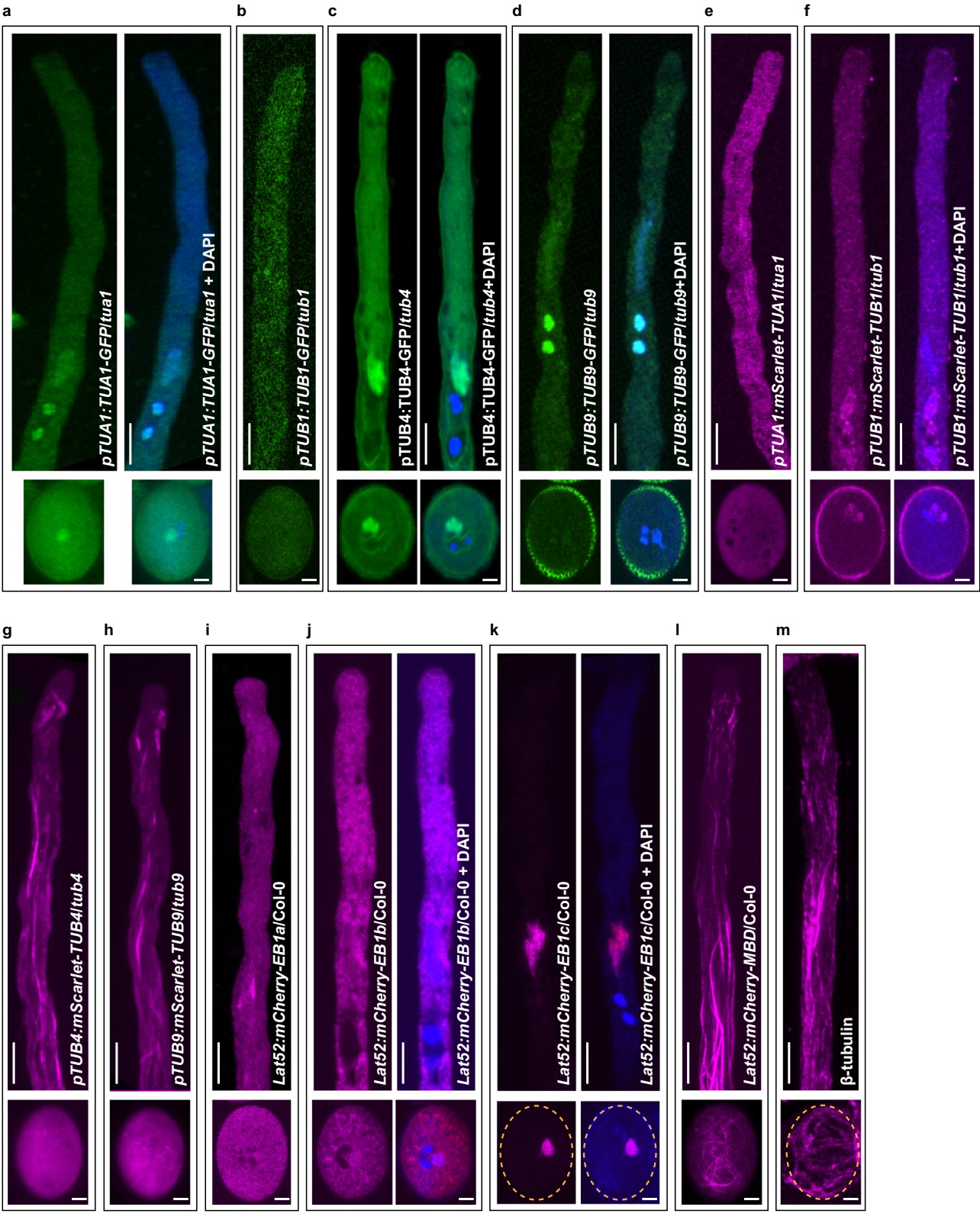

pollen grain to the shank region, and short microtubules were detected in the subapical region, where actin filaments formed the actin fringe (Fig. 2g–i). At the apical dome, while a few fine actin filaments existed, microtubules were hardly observed in these pollen tubes (Fig. 2g, j, k). Altogether, these data indicated that microtubules and actin filaments had distinct initial assembly sites and displayed different structures and dynamics, suggesting that they might play different roles during pollen germination and tube growth.

## Microtubules and VN displayed a spatiotemporal correlation in pollen

We speculated that the sphere-shaped assembly sites of microtubules in pollen grains were around MGU. To test this hypothesis, we

**Fig. 1 | Generation of fluorescently labeled reporter lines for visualizing microtubules in pollen. a–d** Representative images of 15 images showing the subcellular localizations of C-terminal GFP-tagged TUA1, TUB1, TUB4, and TUB9 in pollen grains and tubes. The fusion proteins were expressed under the control of native promoters in the corresponding mutant background, as indicated. The nuclei of VN and SCs in pollen grains and tubes were stained with DAPI. Bar in the upper panel, 10 μm; bar in the lower panel, 5 μm. **e–h** Representative images of 15 images showing the subcellular localizations of N-terminal mScarlet-tagged TUA1, TUB1, TUB4, and TUB9 in pollen grains and tubes. The fusion proteins were expressed under the control of native promoters in the corresponding mutant background, as indicated. The nuclei of VN and SCs in pollen grains and tubes were stained with DAPI. Bar in the upper panel, 10 μm; bar in the lower panel, 5 μm. **i–k** Representative images of 15 images showing the subcellular localizations of N-terminal mCherry-tagged AtEB1a, AtEB1b, and AtEB1c under the control of Lat52 in pollen grains and tubes. The nuclei of VN and SCs in pollen grains and tubes were stained with DAPI. Bar in the upper panel, 10 μm; bar in the lower panel, 5 μm. **l** Representative images of 20 images showing the subcellular localizations of the N-terminal mCherry-tagged microtubule-binding domain of *Arabidopsis* MAP4 (MBD) under the control of Lat52 in pollen grains and tubes. Bar in the upper panel, 10 μm; bar in the lower panel, 5 μm. **m** Representative images of 20 images showing the subcellular localization of microtubules in pollen grains and tubes revealed by indirect immunofluorescence assay using the antibody against β-tubulin. Bar in the upper panel, 10 μm; bar in the lower panel, 5 μm.

examined the spatiotemporal interaction between microtubules and MGU. The widely used marker line Lat52:H2B-GFP labels VN, and DUO1:tdTomato labels SCs[6]. We, therefore, crossed Lat52:H2B-GFP with the mCherry-MBD line, and Lat52:H2B-GFP /DUO1:tdTomato with the mEGFP-MBD line. We observed that microtubules indeed initiated their assembly around VN and SCs during the early stage of hydration (Fig. 3a, b, Supplementary Movie 2). Time-lapse imaging further revealed that VN, wrapped in the microtubule meshwork, underwent a soft act of movement in the opposite direction to the germination site during pollen germination ($n = 30$) (Fig. 3c, d, Supplementary Movie 2). After pollen germination, VN was transported into the pollen tube when the tube reached a mean length of $56.4 \pm 11.8$ μm (mean ± SD, $n = 25$). In fast-growing pollen tubes, VN maintained forward migration interspersed with backward movement or a pause (Fig. 3e, Supplementary Movie 3). Notably, microtubule bundles were always observed to be in front of the trajectory of VN movement, no matter whether the direction was forward or backward (Fig. 3e, Supplementary Movie 3). Detailed observation further revealed that the leading edge of the VN colocalized well with microtubules (Fig. 3e, arrowheads, Supplementary Movie 3), with a duration accounting for $89.0 \pm 7.5\%$ (mean ± SD, $n = 10$) of the total observation time. Fluorescence intensity analysis also supported a spatiotemporal correlation between the leading edge of VN and microtubules (Fig. 3f). The close interaction between microtubules and VN suggested that microtubules may have a role in MGU migration in pollen.

**Microtubules are indispensable for the directional movement of VN in pollen**

To dissect the function of microtubules in MGU migration, we used oryzalin to fully depolymerize microtubules in pollen (Fig. 4a). Oryzalin treatment (1–2.5 μM) led to a significant increase in the pollen germination rate compared to the control (Supplementary Fig. 3a, b), but had a minor impact on the growth rate of pollen tubes (Supplementary Fig. 3c), the velocity of cytoplasmic streaming (Supplementary Fig. 3g) and actin filament organizations (Supplementary Fig. 3i–k), indicating that microtubules are not essential for the tip-growth of pollen tubes. Strikingly, in the absence of microtubules, VN was able to travel to the edge region of pollen grains during pollen germination, resulting in a significantly increased maximal movement distance compared to that of the control (Fig. 4a–c, Supplementary Movie 2). These data suggested that microtubules are important for controlling the movement of VN in pollen grains.

We next applied oryzalin to the dual fluorescent marker line Lat52:H2B-GFP/DUO1:tdTomato. As the MGU migrates in its entirety in pollen, the movement of VN was used to represent the transport of MGU in this study. Time-lapse imaging revealed that VN moved bidirectionally with higher velocity for the forward movement than the backward movement in the control samples (Fig. 4d, f, h). Application of oryzalin resulted in an increased amplitude of both forward and backward movement of the VN, leading to an irregular movement pattern (Fig. 4e, g). VN could not stably stay in the pollen tube after it left the pollen grain in the absence of microtubules (Fig. 4e, g),

implying that microtubules were essential for maintaining the steady movement of VN in pollen tubes. In line with this notion, quantification of the velocity of VN movement revealed that both forward and backward movement was significantly faster in the presence of oryzalin than in the control (Fig. 4h). Moreover, when quantifying the position of VN in pollen tubes, we found that in contrast to the relatively fixed distance of VN to the tip of control tubes ($42.2 \pm 12.0$ μm, mean ± SD, $n = 645$ from 10 cells), the VN floated in the pollen tube after oryzalin treatment, resulting in an increased range of its distance to the pollen tube tip ($96.4 \pm 46.8$ μm, mean ± SD, $n = 485$ from 10 cells) (Fig. 4i). Consistently, the minimal distance of VN to the pollen tip was significantly decreased, and the maximal distance was significantly increased in the pollen tubes treated with oryzalin compared to the control (Fig. 4j). In despite of this, all of VN eventually entered to pollen tube in the presence of oryzalin. Similar irregular movement of VN was observed in the presence of another microtubule inhibitor, nocodazole[14] (Supplementary Fig. 4a-d). In contrast, the velocity of mitochondria trafficking was not significantly affected by oryzalin and nocodazole treatment (Supplementary Fig. 4e). These data suggested that microtubules were required for the fine-positioning of VN in growing pollen tubes.

To further assess the necessity of microtubules in VN transport, we first depolymerized microtubules with oryzalin treatment and then washed the inhibitor out (Fig. 4k, 0 s). During microtubule recovery, we observed that microtubules first reassembled at the cortex of the subapical region of the pollen tubes (Fig. 4k, 1160–1760 s, arrows, Supplementary Movie 4), and gradually extended to the shank region (Fig. 4k, 2960 s). Before the VN contacted the microtubules, it moved back and forth in the pollen (Fig. 4k, 0–2960 s, Supplementary Movie 4, Fig. 4l). Strikingly, as long as the tip of the VN came into contact with microtubules (Fig. 4k, 3360–3680 s, Supplementary Movie 4, arrowheads), it started a steady forward movement and quickly returned to the front position of the pollen tube ($n = 5$) (Fig. 4k, 3360–3680 s, Supplementary Movie 4, Fig. 4l). Taken together, these data support the important roles of microtubules in maintaining the stable localization and directional movement of VN in both pollen grains and tubes.

Mutations of *WIT1* and *WIT2* led to impaired VN movement in pollen tubes[5]. To examine whether the cytoskeleton dynamics was affected in the mutant, we introgressed mCherry-MBD and Lifeact-mEGFP into *wit1 wit2* double mutants, and time-lapse images of the fluorescent markers were collected, respectively. The overall dynamics were quantified via a correlation coefficient analysis[26]. This analysis assessed the extent of cytoskeleton rearrangements over time by calculating the correlation of fluorescent intensity at all pixel positions between all pairwise temporal intervals. The global cytoskeleton dynamics over time were reflected by the rate of decay of correlation coefficient values as the temporal interval increased. The correlation coefficient curves decayed similarly in *wit1 wit2* double mutants as in the control pollen tubes (Supplementary Fig. 5a–d). These data suggested that the transportation defect of VN in *wit1 wit2* double mutants was not likely caused by a defect in cytoskeleton dynamics.

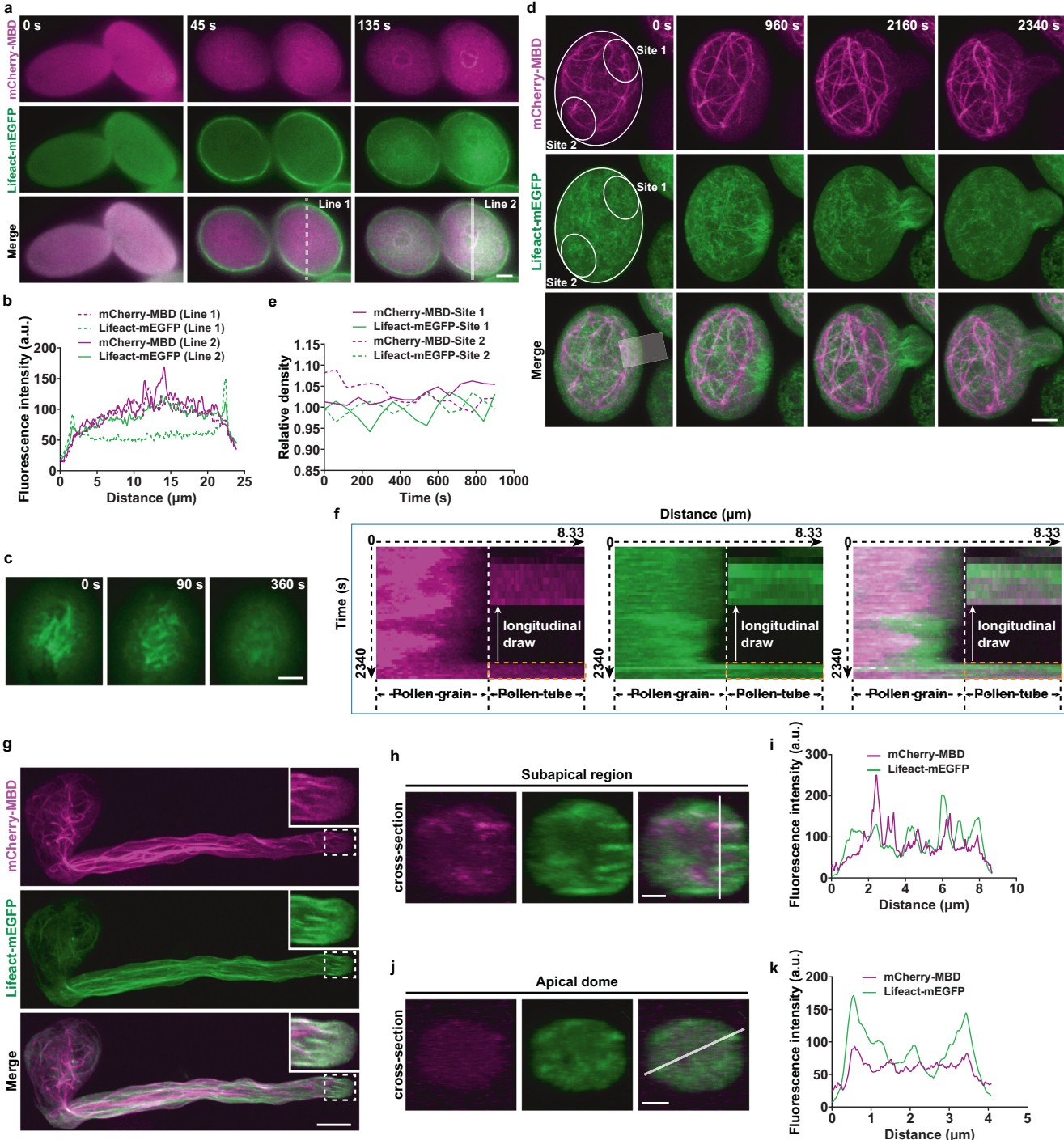

**Fig. 2 | The dynamics of microtubules and actin filaments in pollen. a** Time-lapse images of mCherry-MBD-labeled microtubules and Lifeact-mEGFP-labeled actin filaments shortly after hydration in pollen grains. "0 s", the point when mature pollen was unhydrated. Bar, 5 μm. **b** Fluorescence intensity of mCherry-MBD and Lifeact-mEGFP along the white lines in (**a**). **c** Time-lapse images of Lifeact-mEGFP-labeled actin filaments at cell cortex treated by LatB. Bar, 2 μm. **d** Time-lapse images showing the arrangement and dynamics of microtubules and actin filaments in the germinating pollen grains. Bar, 5 μm. **e** Relative fluorescent density of mCherry-MBD and Lifeact-mEGFP at Sites 1 and 2 in (**d**). **f** Kymograph analysis of mCherry-MBD and Lifeact-mEGFP in the region indicated by the white band in (**d**). The region marked with yellow dotted squares is longitudinally drawn. **g** Representative images of 20 images showing mCherry-MBD and Lifeact-mEGFP in pollen tubes. Bar, 10 μm. **h** Cross-section images of mCherry-MBD and Lifeact-mEGFP at the subapical region of the pollen tube in (**g**). Bars, 2 μm. **i** Fluorescence intensity of mCherry-MBD and Lifeact-mEGFP at the subapical region of the pollen tube along the white line in (**h**). **j** Cross-section images of mCherry-MBD and Lifeact-mEGFP at the apical region of the pollen tube in (**g**). Bars, 2 μm. **k** Fluorescence intensity of mCherry-MBD and Lifeact-mEGFP at the apical region of the pollen tube along the white line in (**j**). Source data are provided as a Source Data file.

## Kinesin is involved in the directional movement of the VN

Kinesins are a superfamily of motors that drive microtubule-based transport in cells. There are many kinesins in pollen, but their functions are largely unknown[23]. In worms, rice, and moss, kinesins have been reported to be required for the nuclear migration process in somatic cells[27–29], raising the possibility that kinesin may also be involved in the microtubule-regulated migration of MGU in pollen. To test this hypothesis, we applied BTB-1, a drug that inhibits the ATPase activity of

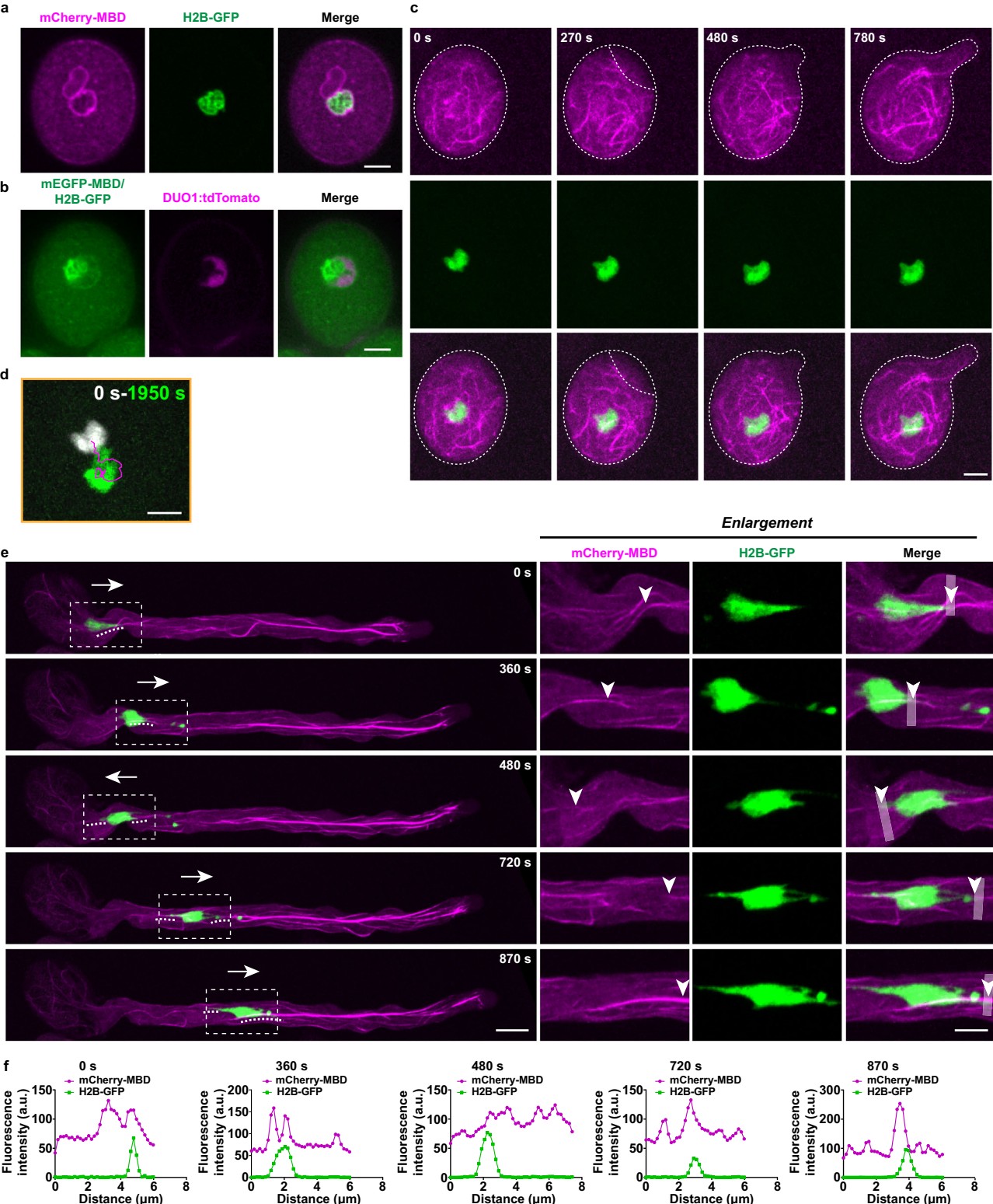

**Fig. 3 | The spatiotemporal interaction of microtubules and VN in pollen grains and tubes. a** Representative images of 15 images showing mCherry-MBD and H2B-GFP in pollen grains during the early stage of hydration. Bar, 5 μm. **b** Representative images of 15 images showing mEGFP-MBD, H2B-GFP, and DUO:tdTomato in pollen grains during the early stage of hydration. The green fluorescence surrounding the DUO:tdTomato-labeled SCs was absent in the H2B-GFP alone pollen grain in (**a** middle panel), indicating that this fluorescence represented the localization of mEGFP-MBD, but not H2B-GFP. Bar, 5 μm. **c** Time-lapse images showing the movement of microtubules and VN in germinating pollen grains. "0 s" was set at 10 min after the incubation of pollen grains on the germination medium. Bar, 5 μm. **d** Trajectory analysis of the movement of VN along the magenta line in (**c**). Bar, 5 μm. **e** Time-lapse imaging showing the interaction between microtubules and VN in growing pollen tubes. The regions marked with white dotted squares are enlarged in the right panels. The contact sites of microtubules and the leading edge of the VN are indicated with arrowheads. Bar in the left panel, 10 μm; bar in the right panel, 5 μm. **f** Fluorescence intensity of mCherry-MBD and H2B-GFP is perpendicular to the leading edge of the VN along the white line in (**e**). Source data are provided as a Source Data file.

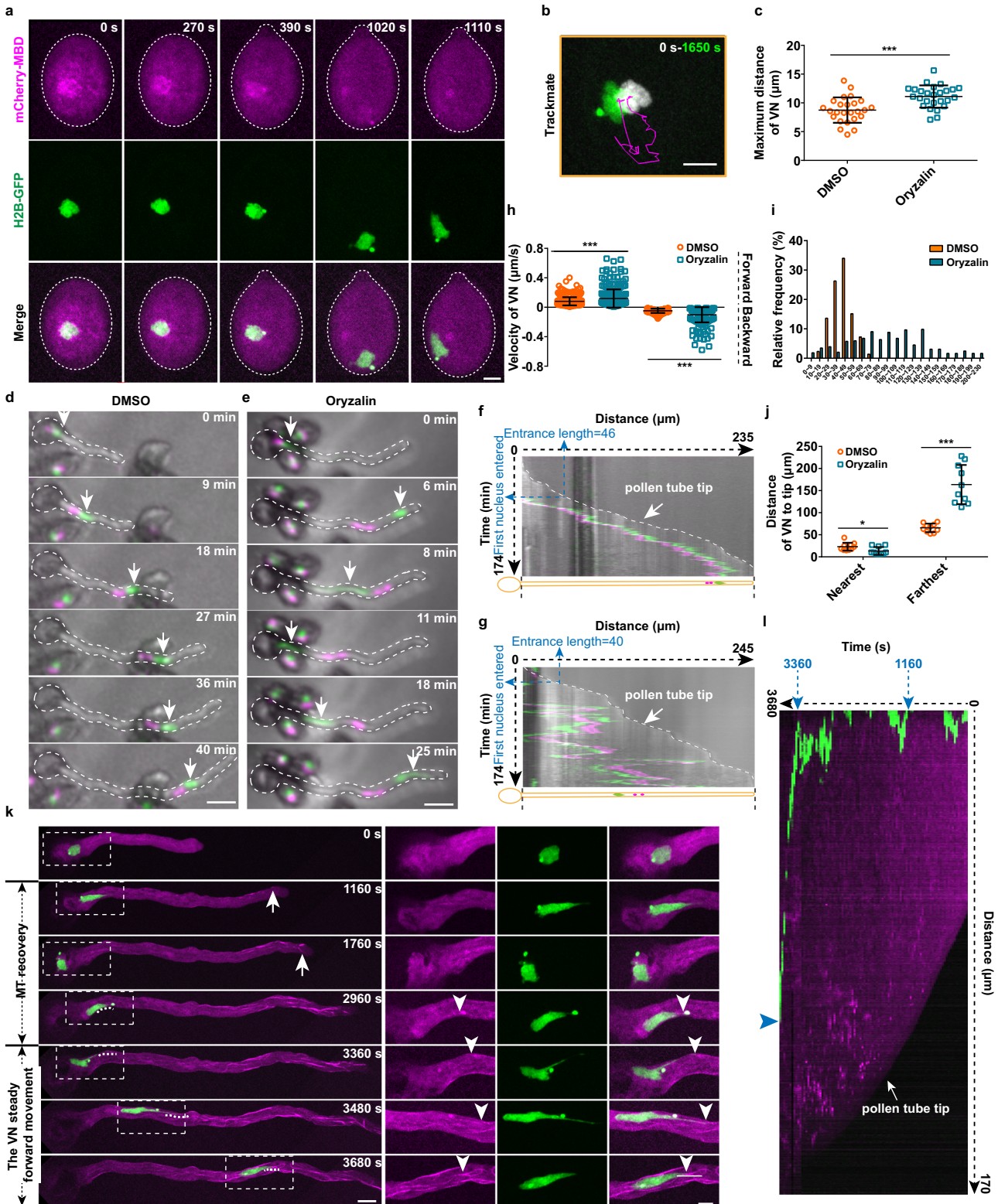

the motor domain of a kinesin[30,31], to growing pollen tubes. Time-lapse imaging and kymograph analysis showed that, compared with the smooth mode of VN movement in the control (Fig. 5a, c), BTB-1 treatment caused phenotypes of VN migration similar to those of oryzalin treatment (Fig. 5b, d), including an increased speed of VN movement (Fig. 5e) and an elevated distance of VN to the pollen tube tip (Fig. 5f). Similar to oryzalin treatment, while the pollen germination rate was significantly increased (Supplementary Fig. 3d, e), the growth

rate of pollen tubes (Supplementary Fig. 3d, f), the velocity of vesicle trafficking (Supplementary Fig, 3h), actin filament organizations (Supplementary Fig. 3i–k) nor microtubule dynamics (Supplementary Fig. 5e, f) were significantly affected by BTB-1 treatment. To strengthen the data from pharmacological perturbation, we attempted to identify kinesins involved in MGU transportation. *KIN14G* and *KIN14H* were two kinesins belonging to the same subclade in the phylogenetic tree[32] and were preferentially expressed in mature pollen (Supplementary

**Fig. 4 | The stable directional movement of VN in pollen relies on microtubules.**
**a** Time-lapse images of mCherry-MBD and H2B-GFP in germinating pollen grains upon the application of oryzalin. "0 s", 10 min after the incubation of pollen grains on a medium containing 1.5 µM oryzalin. Bar, 5 µm. **b** Trajectory analysis of VN movement in (**a**). Bar, 5 µm. **c** Quantification of the maximum distance of VN movement in pollen grains without ($n = 25$) or with ($n = 26$) oryzalin treatment. **d**, **e** Time-lapse images of VN (green, arrows) and SCs (magenta) in DMSO (**d**) or oryzalin-treated growing pollen tubes (**e**). Bar, 30 µm. **f**, **g** Kymograph analysis of (**d**) and (**e**). The pollen tube tip is indicated with an arrow. **h** Quantification of the velocity of VN movement in DMSO and oryzalin-treated pollen tubes. Velocity of VN forward movement, DMSO ($n = 494$), Oryzalin ($n = 270$); velocity of VN backward

movement, DMSO ($n = 148$), Oryzalin ($n = 229$), $n$ from 10 cells for each treatment. **i** The frequency distribution of the distance of VN to the tip of DMSO and oryzalin-treated pollen tubes. **j** Quantification of the minimum and maximum distance of VN to the tip of DMSO and oryzalin-treated pollen tubes. $n = 10$ pollen tubes for each treatment. **k** Time-lapse images of mCherry-MBD and H2B-GFP in pollen tubes after oryzalin washout. The region marked with white dotted squares is enlarged in the right panels. The reassembly sites of microtubules are indicated by arrows, and the contact sites of VN tips and microtubules are pointed out with arrowheads. $n = 5$. Bar in the left panel, 10 µm; bar in the right panel, 5 µm. **l** Kymograph analysis of (**k**). Values are mean ± SD; * $P < 0.05$,*** $P < 0.001$; two-tailed Student's $t$-test in (**c**), (**h**) and (**j**). Source data are provided as a Source Data file.

Fig. 6a, b). To examine their potential function, we obtained T-DNA insertional mutants of *kin14g* and *kin14h*, respectively. Given that *KIN14G* and *KIN14H* might work redundantly, we crossed *kin14g* and *kin14h* to obtain the *kin14g kin14h* double mutant. The VN moved in an irregular way, with significantly increased velocity for both forward and backward movement, in the double mutants (Fig. 5g–k; Supplementary Fig. 6c, d). The distance of VN to the pollen tip was also markedly increased in *kin14g kin14h* pollen tubes compared to the control (Fig. 5l). These observations were consistent with those observed in the pollen tubes treated with kinesin inhibitors BTB-1. Together, these data suggested that kinesin plays an important role in the directional transport of VN in pollen tubes.

To further dissect the function of kinesin in VN migration along microtubules, we first depolymerized microtubules and inhibited kinesin activity simultaneously through oryzalin and BTB-1 co-treatment, and then only restored microtubules by washing out oryzalin (Fig. 5m, 0 s). Similar to the recovery process after microtubule depolymerization, the microtubules reassembled and then gradually extended to the shank region of the pollen tube (Fig. 5m, 320–2000 s, arrow, Supplementary Movie 5). At this time, the movement of VN was unstable, moving forwards or backward between the pollen grain and tube (Fig. 5m, Supplementary Movie 5, Fig. 5n). When the leading edge of the VN touched the cortical microtubules extending to the base of the pollen tube (Fig. 5m, 2000–2760 s, arrowheads, Supplementary Movie 5), it began to move steadily towards the tip of the pollen tube (Fig. 5m, 2000–2760 s, Supplementary Movie 5, Fig. 5n). Notably, the VN became significantly longer ($33.9 ± 9.5$ µm, mean ± SD, $n = 35$) (Fig. 5m, 2360, 2520 s, Supplementary Movie 5) in the presence of BTB-1 compared to the control ($24.8 ± 6.8$ µm, mean ± SD, $n = 35$). Subsequently, although the VN was able to return to the front of the pollen tube, its motion remained unstable and oscillatory (Fig. 5m, n). Together, these data provide evidence that kinesin may coordinately function with microtubules by finely modulating the directional movement of MGU in pollen tubes.

### Cytoplasmic streaming impacts VN migration in pollen

The actomyosin system is generally recognized to be responsible for cytoplasmic streaming by providing the directional driving force that allows the movement of organelles and vesicles along actin filaments[33,34]. We speculated that the cytoplasmic streaming might account for the increased speed of VN movement after the depolymerization of microtubules or inhibition of kinesin activity in pollen (Figs. 4h and 5e, k). To test this hypothesis, we first examined the possible correlation between VN movements and the cytoplasmic streaming surrounding the VN in terms of their directions and velocities (Fig. 6). No matter during the forward or backward movement of the VN, the direction of the cytoplasmic streaming surrounding VN was quite inconsistent with the migration direction of the VN (Fig. 6a, c, Supplementary Movie 6). However, during the pause period of the VN, the cytoplasmic streaming had no major direction (Fig. 6b, Supplementary Movie 6). Consistently, quantification of the anisotropy of vesicle trafficking direction revealed a significantly reduced value when VN was paused compared to the values for mobile VN (Fig. 6d).

The correlation between VN movement and the cytoplasmic streaming implies a possible role of the cytoplasmic streaming in VN migration.

To investigate the function of the actomyosin system in VN movement, the Lifeact-mCherry line was crossed with the H2B-GFP line, and LatB and PBP[35], a drug inhibiting the activity of myosin, were applied, respectively. Time-lapse imaging found that VN moved backward during pollen germination (Fig. 6e, Supplementary Movie 7). Trajectory analysis revealed that the backward movement of VN was markedly inhibited upon LatB and PBP treatment compared to the control (Fig. 6f, g). As both LatB and PBP treatment led to severe defects in pollen germination, we applied the drugs one hour later after pollen was cultured on a normal medium to allow germination and examined their effect in pollen tubes. After the LatB treatment, the pollen tube stopped growing rapidly, and VN gradually moved to near the tip of the pollen tube and then stopped (Fig. 6h, i). However, after PBP treatment, the pollen tube could still grow, and actin filament dynamics were not affected (Supplementary Fig. 5g, h), but the migration rate of VN slowed down significantly (Fig. 6h, i). Moreover, with the increase of PBP concentration, the velocity of cytoplasmic streaming gradually decreased, and the velocity of VN also decreased correspondingly (Fig. 6j, k). To further evaluate the effect of cytoplasmic streaming on VN movement, we obtained single and double mutants of two myosins, *Myo11C1* and *Myo11C2*, which were highly expressed in mature pollen[36] (Supplementary Fig. 6e, f). The velocity of cytoplasmic streaming and the VN were significantly slowed down in *myo11c1*, *myo11c2*, and *myo11c1 myo11c2* pollen tubes compared to the control (Fig. 6l, m). The VN was significantly closer to the tip in the single and double mutants than in the control pollen tubes (Fig. 6n; Supplementary Fig. 6g–j). These observations mimicked those observed in the pollen tubes treated with myosin inhibitors PBP. Detailed observations of pollen tubes showed that, in contrast to continuous contacts with microtubules (Fig. 3e), the duration of contacts between the leading edge of VN and actin filaments largely decreased to be $61.1 ± 5.9\%$ (mean ± SD, $n = 10$) of the total observation time (Supplementary Fig. 7, Supplementary Movie 8). Altogether, these results suggested that the cytoplasmic streaming had a significant impact on the velocity of MGU oscillational migration in pollen.

### Microtubules participate in coordinating the directional movement of MGU

To further study the functions of the cytoskeleton in regulating MGU movement, a series of washout experiments were performed on growing pollen tubes. Pollen tubes were first treated with a high dose of LatB and then transferred to LatB-free media to thoroughly wash away the inhibitor. We surprisingly observed that some pollen tubes re-established their polarity: the original tube stopped growing, and a new tube branched out (Supplementary Fig. 8a). Interestingly, the MGU returned to the branching site from the original tube and migrated into the new tube (Supplementary Fig. 8a, b). These observations suggested that MGU closely pursued the tip-growth of the pollen tube. However, when both actin filaments and microtubules were depolymerized by simultaneous treatment with LatB and oryzalin

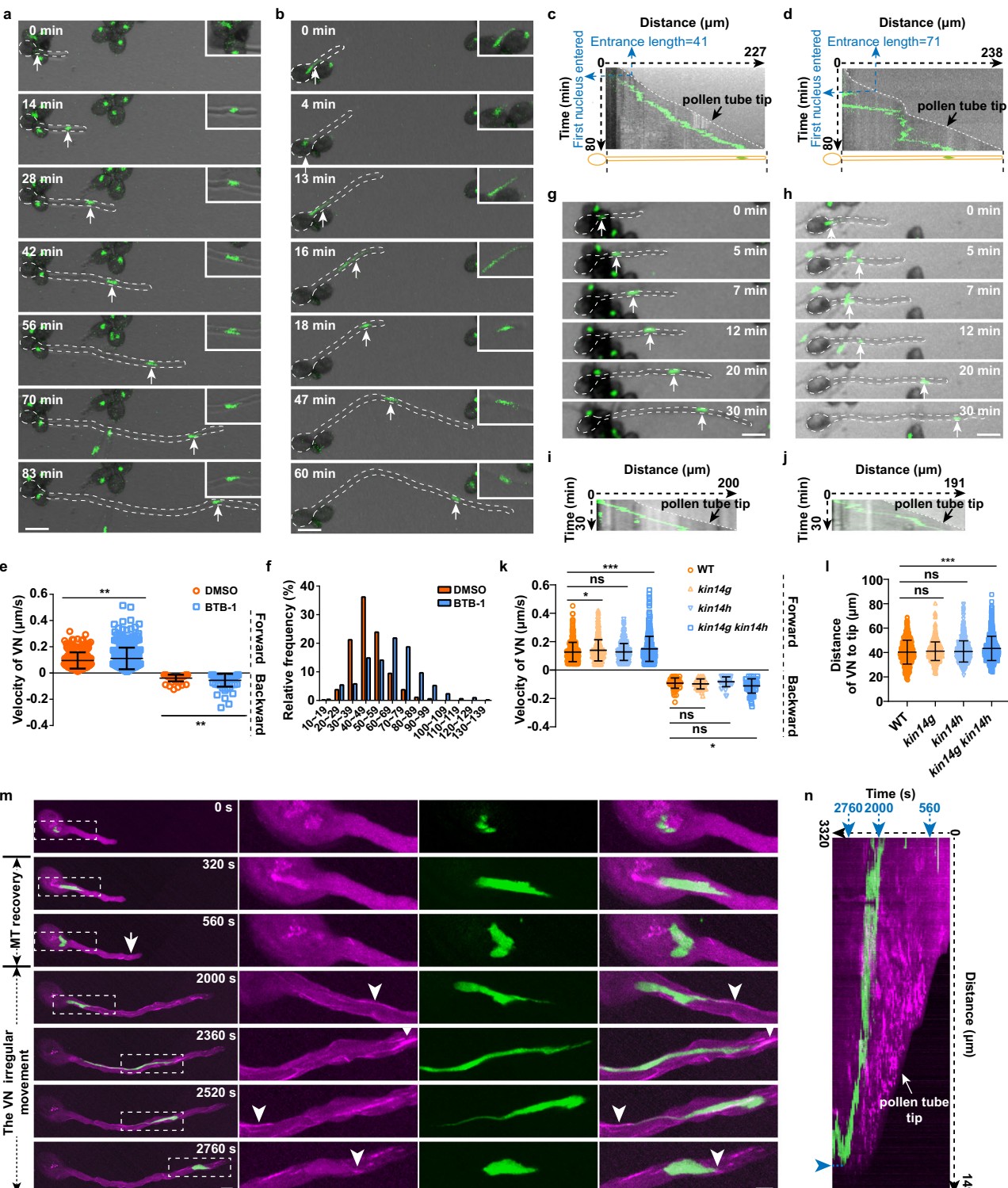

**Fig. 5 | Kinesin is involved in the directional movement of VN. a**, **b** Time-lapse images of H2B-GFP in DMSO- (**a**) or BTB-1-treated growing pollen tubes (**b**). Images were enlarged to show VN morphology. Bar, 30 µm. **c**, **d** Kymograph analysis of (**a**) and (**b**). **e** Quantification of VN's movement velocity in DMSO and BTB-1 treatment. Velocity of VN forward movement, DMSO (*n* = 310), BTB-1 (*n* = 342); velocity of VN backward movement, DMSO (*n* = 63), BTB-1 (*n* = 78), *n* from 10 cells for each treatment. **f** The frequency distribution of the distance of VN to the tip of DMSO and BTB-1 treatment. **g**, **h** Time-lapse images of H2B-GFP in WT (**g**) or *kin14g kin14h* double mutant growing pollen tubes (**h**). Bar, 30 µm. **i**, **j** Kymograph analysis of (**g**) and (**h**). **k** Quantification of VN's movement velocity in WT and kinesin mutants. Velocity of VN forward movement, WT (*n* = 402), *kin14g* (*n* = 393), *kin14h* (*n* = 416) from 10 cells for each genotype, *kin14g kin14h* (*n* = 521) from 14 cells. Velocity of VN

backward movement, WT (*n* = 66), *kin14g* (*n* = 49), *kin14h* (*n* = 45) from 10 cells for each genotype, *kin14g kin14h* (*n* = 66) from 14 cells. **l** The distance of the VN to the tip of WT and kinesin mutants. WT (*n* = 795), *kin14g* (*n* = 645), *kin14h* (*n* = 778), *kin14g kin14h* (*n* = 745) from 10 cells for each genotype. **m** Time-lapse images of mCherry-MBD and H2B-GFP in pollen tubes after oryzalin and BTB-1 co-treatment and followed by oryzalin washout. The region marked with white dotted squares is enlarged in the right panels. Arrow indicates the reassembly site of microtubules. Arrowheads indicate the contact sites of VN tips and microtubules. Bar in the left panel, 10 µm; bar in the right panel, 5 µm. **n** Kymograph analysis of (**m**). Values are mean ± SD; *P < 0.05, **P < 0.01, ***P < 0.001, ns, not significant; two-tailed Student's *t*-test in (**e**), (**k**), (**l**). Source data are provided as a Source Data file.

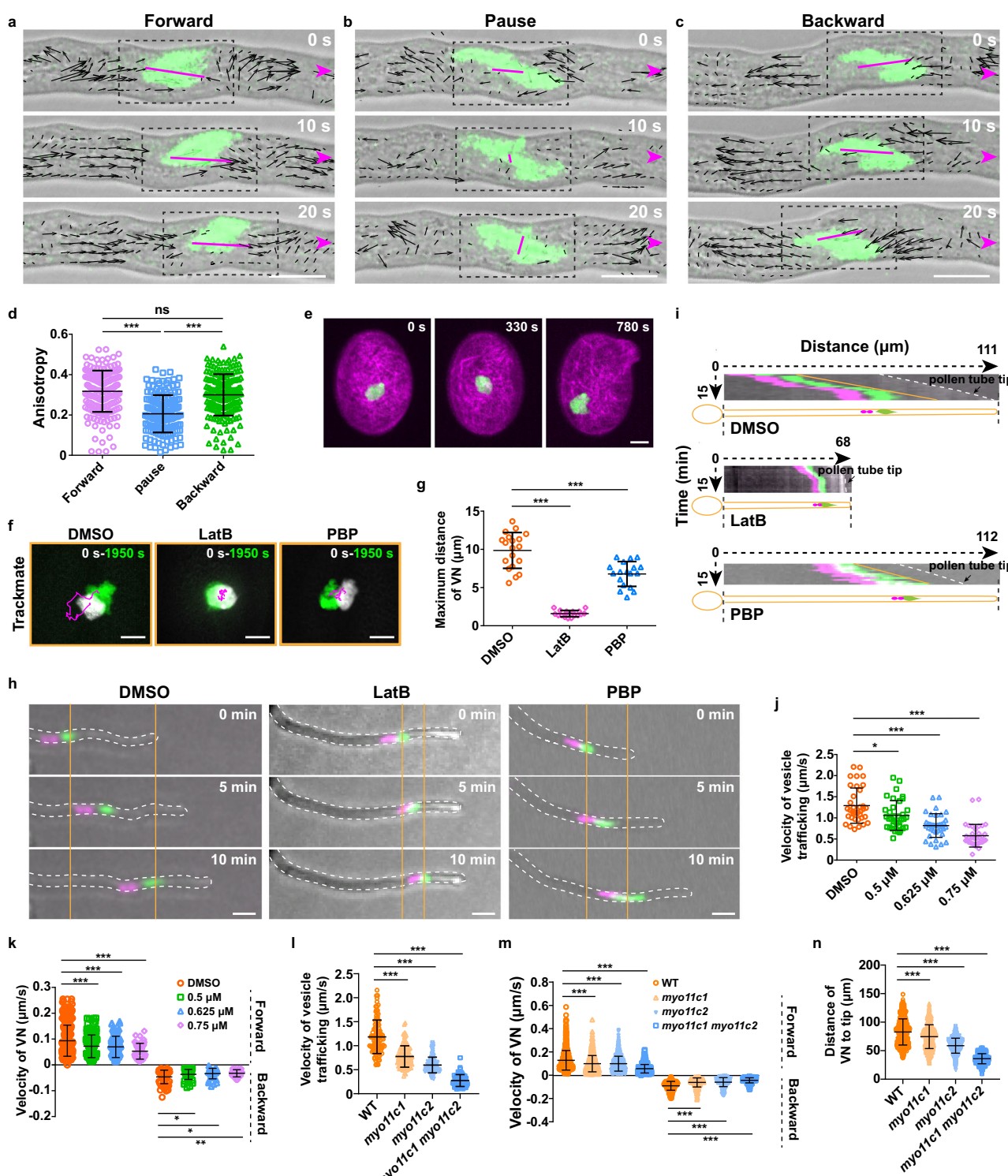

and only LatB was washed out, the pattern of MGU migration was markedly altered (Supplementary Fig. 8c, d). In this case, 50% ($n = 20$) of MGUs failed to enter the branched pollen tube (Supplementary Fig. 8c). These MGUs hesitated at the branching site of the pollen tube, and the VNs underwent several trials between the new and old pollen tube tips but did not enter either the new or the old pollen tube tip. Instead, they oscillated repeatedly in the original pollen tube (Supplementary Fig. 8c, e, f), resulting in a significantly increased distance between the VN and the new pollen tube tip compared to the control (LatB washout after LatB treatment; Supplementary Fig. 8i). In the rest 50% of pollen tubes, the MGUs successfully entered the new pollen

tube, but their motion modes were still oscillating (Supplementary Fig. 8d, e, g). We then washed out oryzalin after treatment with both LatB and oryzalin. Neither pollen tube tip-growth nor MGU movement was restored (Supplementary Fig. 8h, i). These data corroborated that MGU movement correlated with pollen tube growth and microtubules were essential for MGUs to follow and keep pace with the tip growth of pollen tubes.

## Discussion
In this study, we provide a full picture of the dynamic interplay among microtubules, actin filaments, and MGUs in both pollen grains and

**Fig. 6 | Cytoplasmic streaming impacts on VN movement in pollen tubes. a–c** Time-lapse images of H2B-GFP in pollen tubes when VN moves forward (**a**), backward (**b**), or at pause (**c**). Black arrows indicate the vector distribution of the velocity of vesicles. Arrowheads indicate the tube growth direction. Magenta straight lines indicate the anisotropy of vesicle trafficking. Bars, 10 µm. **d** Quantification of the anisotropy of vesicle trafficking. Forward ($n = 186$), Pause ($n = 154$), and Backward ($n = 177$) from 10 cells. **e** Time-lapse images of Lifeact-mCherry and H2B-GFP in the germinating pollen grains. "0 s" sets at 10 min after the incubation on medium. Bar, 5 µm. **f** Trajectory analysis of VN movement in DMSO-, LatB- or PBP-treated pollen grains. Bars, 5 µm. **g** Quantification of the maximum distance of VN movement in DMSO, LatB, or PBP treatment. DMSO ($n = 19$), LatB ($n = 18$) and PBP ($n = 18$). **h** Time-lapse images of VN (green) and SCs (magenta) in DMSO-, LatB- or PBP-treated pollen tubes. Bars, 20 µm. **i** Kymograph analysis of (**h**). **j** Quantification of the velocity of vesicle trafficking in DMSO and PBP treatment. $n = 36$. **k** Quantification of VN's movement velocity in DMSO and PBP treatment.

Velocity of VN forward movement, WT ($n = 310$), 0.5 µM PBP ($n = 135$), 0.625 µM PBP ($n = 116$), 0.75 µM PBP ($n = 44$). Velocity of VN backward movement, WT ($n = 63$), 0.5 µM PBP ($n = 56$), 0.625 µM PBP ($n = 34$), 0.75 µM PBP ($n = 22$). **l** Quantification of the velocity of vesicle trafficking in WT and myosin mutants pollen tubes. WT ($n = 104$), *myo11c1* ($n = 113$), *myo11c2* ($n = 103$), *myo11c1 myo11c2* ($n = 103$). **m** Quantification of VN's movement velocity in WT and myosin mutants pollen tubes. Velocity of VN forward movement, WT ($n = 704$), *myo11c1* ($n = 691$), *myo11c2* ($n = 735$), *myo11c1 myo11c2* ($n = 690$).Velocity of VN backward movement, WT ($n = 170$), *myo11c1* ($n = 140$), *myo11c2* ($n = 107$), *myo11c1 myo11c2* ($n = 134$). **n** Quantification of the distance of VN to the tip of WT and myosin mutants. WT ($n = 792$), *myo11c1* ($n = 812$), *myo11c2* ($n = 863$), *myo11c1 myo11c2* ($n = 800$). *n* from 10 cells for each treatment in (**j**), (**k**); *n* from 10 cells for each genotype in (**l**), (**m**), (**n**). Values are mean ± SD; *$P < 0.05$, **$P < 0.01$, ***$P < 0.001$, ns, not significant; two-tailed Student's *t*-test in (**d**), (**g**), (**j–n**). Source data are provided as a Source Data file.

tubes. Based on our data, we propose that while the actomyosin system-based cytoplasmic streaming impacts on the velocity and direction of local VN movement in its pollen-tube journey, the microtubule network is key for the fine-positioning of VN in growing pollen tubes and for the VN to follow and keep pace with the tip growth of pollen tubes (Supplementary Fig. 9).

Due to the lack of a live-cell imaging tool for visualizing microtubules in pollen, microtubule dynamics and their spatiotemporal interactions with MGUs, as well as with actin filaments, have been largely unknown. A previous attempt to visualize microtubules in living pollen tubes with Lat52-driven GFP-TUA1 and GFP-TUB6 failed[14], suggesting that in vivo labeling of microtubules in pollen is challenging. Our efforts to check the subcellular localization of fluorescently tagged tubulins and microtubule-associated proteins detected different labeling patterns of microtubules in pollen grains and tubes, such as accumulated signals around VN and/or SC or filamentous structures in the cytoplasm (Fig. 1). Thus, different microtubule markers may achieve more pronounced views on certain microtubule subpopulations[13]. The transient transformation of GFP-AtEB1c detected extensive microtubule bundles in tobacco pollen tubes[13]; in contrast, mCherry-AtEB1c in our study was specifically labeled VN. This might be because the nuclear localization signal at the C-terminus of AtEB1c leads the fusion protein to VN in *Arabidopsis* but may lose its function in tobacco. In our study, mCherry-MBD was the only marker able to label microtubules both in pollen grains and tubes, providing a useful tool for analyzing the dynamic interaction among microtubules, actin filaments, and MGUs during the overall process from pollen germination to tube elongation.

It is well-established that the actomyosin network generates the driving force for the long-distance vesicle and organelle transport in pollen tubes[2,15–18]. The directional movement of vesicles and organelles in pollen tubes generates very fast cytoplasmic streaming, from the grain to the tip along the cortex and from the tip to the base in the center of the tube[37]. We observed that the motility of the VN was an order of magnitude lower than the mean velocity of cytoplasmic streaming in pollen tubes. Hence, maintaining a consistent pace with the growing pollen tube and not drifting with cytoplasmic streaming is a challenging task for MGU. We detected long-term contact of the VN tip with microtubules in the pollen tube, but a decreased contacting time with actin filaments, suggesting that microtubules may serve as tracks for VN directional transport. We speculate that the intermittent interaction with actin filaments of VN resembles a hitchhiking pathway through the cytoplasmic circulation. Supporting this notion, the irregular and magnified oscillating motion of VN after oryzalin treatment mimics passive movement along with cytoplasmic streaming, during which the direction of MGU travel might depend on whatever the direction of the cytoplasmic circulation that it encounters, consistent with our observations the correlation between VN movement and the cytoplasmic streaming (Fig. 6a–c). However, as immunolocalization of

myosin I and an in vivo marker of F-actin showed pronounced fluorescent signals around VN in pollen[21,38], the direct contribution of actomyosin in promoting MGU transport could not be excluded. Possibility also exists that the VN leading edge is a result of interaction with actin filaments running parallel to the microtubules. Further studies of simultaneously visualize actin filaments, microtubules, and VN dynamics are needed to fully uncover the functional connection and interaction between the two cytoskeletons and the VN.

Our results showed that the initial assembly of microtubules tends to occur around MGU, whereas actin filaments tend to polymerize near the plasma membrane shortly after pollen hydration. The actin filaments near the plasma membrane may modulate endocytosis and exocytosis and regulate $Ca^{2+}$ channel activity, which is crucial for pollen germination[2,39,40]. During germination, the cage-like meshwork of microtubules keeps MGU in the center of pollen grains, while actin filaments rotate on the edge[19]. The distinct loading pattern of these two cytoskeletons suggested that microtubules may serve as a physical constraint for MGU in pollen grains, thus protecting them from rotation with the cytoplasmic circulation driven by actin filaments. When the actin filaments formed a collar-like structure at the germination site, the microtubules and VN shifted to the opposite side of the germination site. Considering the reported role of the actin cytoskeleton in regulating the selection of pollen germination sites[19], we proposed that the retreat of microtubules may create enough space for the establishment of pollen polarity mediated by actin filaments or may only be a result of the squeezing of the construction of the actin structure. In line with these results, oryzalin-induced microtubule depolymerization resulted in a larger amplitude of VN motion in both pollen grains and pollen tubes and microtubule recovery led to a rapid return of the VN to its proper position in pollen tubes and subsequent small concussional migration. These data revealed that the dynamic interaction between microtubules and MGU is required for the homeostatic movement of MGU in both pollen grains and pollen tubes. The fluorescence of mCherry-MBD and mEGFP-MBD around the MGU during the early stage of hydration was very weak. When the cytosolic microtubules were assembled, the fluorescence of these microtubules was strong, which challenged the observation of microtubule signals around the MGU. We thus could not rule out the possibility that some microtubules may form from the MGU during pollen tube elongation. The VN morphology was changed when microtubules were disassembled. It is possible that depolymerization of microtubules impaired the coordination between the movement of SCs and the VN. When the direction or velocity between the SCs and the VN are not matched, the SCs will drag the VN to deform it. We also found that the migration velocity of VN was much lower than that of cytoplasmic streaming in pollen tubes (Figs. 4h, 6j), suggesting that the microtubule system is the speed controller for VN movement in the growing pollen tube. A relatively stable position of MGU near the tube tip, on the one hand, may be essential for the process of discharging SCs when

the pollen tube tip recognizes the egg apparatus and ruptures[41] and may be linked to VN's transcriptional activity regulating communications with the female[5].

It has been reported to that KCBP and OsKCH1, kinesins in moss and rice, respectively, are required for nuclear migration in somatic cells[27,29]. In worms and mammals, kinesin-1 was found to contribute to microtubule-dependent nuclear migration[28,42,43]. Moreover, the authors also revealed that KASH proteins interacted with kinesin-1 and acted as nuclear-specific cargo adaptors for kinesin-1-mediated nuclear migration[28,43]. KASH proteins are the outer nuclear membrane-localized members of the linkers of the nucleoskeleton and the cytoskeleton (LINC) complexes that transduce cytoskeletal forces to the nucleus[44]. The essential roles of the kinesin-LINC module in regulating nuclear migration in both plants and animals suggest that the module may participate in MGU movement during pollen tube growth. Supporting this hypothesis, WIT, and WIP, two *Arabidopsis* KASH proteins, are indispensable for normal VN migration in pollen tubes[5]. We have shown that the impaired VN transport in *wit1 wit2* double mutants is not likely resulted from a defect in cytoskeleton dynamics. WIT proteins can physically interact with myosin proteins[45]. It is possible that the connection of VN with the cytoskeleton is attenuated in the *wit1 wit2* pollen tubes. KCH, a kinesin family member that can move along both microtubules and actin filaments, was assumed to be responsible for the saltatory movement of SCs in *Arabidopsis* pollen tubes based on its specific properties derived from in vitro experiments[22]. Here we have shown the involvement of two KCH isoforms, KIN14G and KIN14H, in microtubule-regulated MGU migration. Despite of generally consistent phenotypes, the MGU migration defect in *kin14g kin14h* double mutants is moderate compared to those observed in pollen tubes treated with microtubule or kinesin inhibitors. These data suggest that other kinesin proteins may also contribute to MGU migration. Detailed analysis of the functions of KIN14G and KIN14H, as well as other kinesins involved, is required to provide mechanistic insight into the complex process of MGU transport in pollen tubes.

The mechanism we proposed for controlling MGU mobility resembles the restricted effect of microtubules on organelle transport in pollen tubes, where microtubules and kinesins are proposed to slow down the local trafficking and tune the final positioning of some organelles, e.g., Golgi bodies and mitochondria after they have been rapidly transported along actin filaments[23,46]. In vitro, motility assays showed that the movement of mitochondria and Golgi was slow and continuous along microtubules compared with the fast and irregular motility along actin filaments, and the addition of microtubules to actin filaments turned organelles to use lower velocities[46]. These data suggest that there may be a common mechanism underlying organelle and MGU migration in pollen, which are all influenced by the cytoplasmic streaming but are finely modulated by the microtubule network.

In conclusion, this study provides new insights into the sophisticated delivery system of the MGU in flowering plants at the live-cell level. Further biochemical and genetic approaches would provide answers to remaining questions regarding key factors responsible for MGU movement and microtubule-actin filament crosstalk.

## Methods
### Plant materials and growth conditions
*Arabidopsis thaliana* ecotype Columbia-0 (Col-0) was taken as the wild type. The T-DNA insertional mutants, *tua1* (SALK_097254), *tub1* (SALK_036755), *tub4* (SALK_204506), *tub9* (SALK_015876C), *kin14g* (SALK_133582), *kin14h* (SALK_106474), *myo11c1* (SALK_129231C), *myo11c2* (SALK_089338C), *wit1* (GABI_470E06) and *wit2* (SALK_127765) were obtained from Nottingham Arabidopsis Stock Centre (NASC) (http://arabidopsis.info/). *kin14g* and *kin14h* were crossed to generate the double mutant. *myo11c1 myo11c2* double mutant was kindly shared

by Prof. Andreas Nebenführ[36]. *wit1 wit2* double mutant was generated by crossing *wit1* and *wit2*. All the primers for genotyping are listed in Supplementary Table 2. The fluorescent plants of ProLAT52:H2B-GFP/ ProDUO1:tdTomato and Lat52:Lifeact-mEGFP were described previously[6,19]. Seeds were stratified for 3 days at 4 °C after surface sterilization and then grown on vertical plates containing half-strength Murashige and Skoog (MS) medium supplemented with 1% (w/v) agar at pH 5.8. Seven-day-old seedlings were transferred to mixed soil and cultured under long-day conditions (16 h light/8 h dark cycles) at 22 °C with a light intensity of 120 µmol/m²/s using LED bulbs.

### Plasmid construction and plant transformation
To generate the TUA1-GFP, TUA4-GFP, TUB1-GFP, TUB4-GFP, and TUB9-GFP fusion constructs, the nucleotide sequences containing the native promoter and genomic region were amplified from *Arabidopsis* genomic DNA with the primer pairs *pTUA1-TUA1F/R*, *pTUB1-TUB1F/R*, *pTUB4-TUB4F/R*, and *pTUB9-TUB9F/R*, respectively (Supplementary Table 2). The cloned fusion sequence *pTUA1:TUA1* and *pTUB1:TUB1* were inserted into *pCAMBIA1305* carrying GFP between the restriction enzyme sites of *EcoRI/KpnI* to generate the complementation vectors *pTUA1:TUA1-GFP-pCAMBIA1305* and *pTUB1:TUB1-GFP-pCAMBIA1305*. The cloned fusion sequence *pTUB4:TUB4* was inserted into *pCAMBIA1305 carrying GFP* between the restriction enzyme sites of *EcoRI/SacI* to generate *pTUB4:TUB4-GFP-pCAMBIA1305* vector. The PCR product *pTUB9:TUB9* was inserted into *pCAMBIA1305* carrying GFP between the restriction enzyme sites of *SacI/BamHI* to generate *pTUB9:TUB9-GFP-pCAMBIA1305* vector. Through a homologous recombination method, the promoter sequence, the coding sequence of a fluorescent protein, and the genomic sequences of a gene were sequentially inserted into the *pCAMBIA1300* vector after being digested with the restriction enzymes *EcoRI/HindIII* to generate the following constructs: *pTUA1:mScarlet-TUA1*, *pTUA4:mScarlet-TUA4*, *pTUB1:mScarlet-TUB1*, *pTUB4:mScarlet-TUB4,* and *pTUB9:mScarlet-TUB9*, respectively. All the error-free constructs were transformed into the corresponding mutant background, respectively.

For generating the transgenic plants *Lat52:mCherry-EB1a*, *Lat52:mCherry-EB1b*, *Lat52:mCherry-EB1c*, *Lat52:mCherry-MBD,* and *Lat52:mEGFP-MBD*, the coding sequences of EB1a, EB1b, EB1c, and MBD were amplified from the Col-0 cDNA via the primer pairs *EB1aF/R*, *EB1bF/R*, *EB1cF/R*, and *MBDF/R* fused with mCherry or mEGFP and the Lat52 promoter, respectively. mCherry-EB1a, mCherry-EB1b, mCherry-EB1c, mCherry-MBD and mEGFP-MBD were cloned into the *pCAMBIA1300* vector. These constructs were transformed into the Col-0 background via *Agrobacterium*-mediated transformation using the floral dip method. Transgenic plants were screened for lines with antibiotic resistance and fluorescence intensity. All the primers for cloning are listed in Supplementary Table 2.

### Pollen germination conditions and pharmacological treatments
*Arabidopsis* pollen was obtained from newly opened flowers of 4- to 6-weeks-old plants, and placed on pollen germination medium (PGM, 5 mM CaCl₂, 5 mM KCl, 1 mM MgSO₄, 0.01% (w/v) H₃BO₃, 10% (w/v) sucrose, and 0.15% agar, pH 7.5) at 28 °C under moist conditions. To observe the overall process from hydration to germination, dry pollen grains were firstly smeared on an observing dish supplied with a damp paper to keep moisture, and then image collection was started under an Andor Dragonfly confocal microscope system (488/561 nm laser; ×60 oil objective, N.A. 1.42) equipped with a Zyla 4.2 sCMOS camera. During image capturing, a small amount of liquid medium containing 5 mg/mL agarose was quickly added to the dry pollen grains and initiated pollen hydration. To depolymerize actin filaments at the cell cortex, PGM (5 mg/mL agarose) containing 100 nM LatB was applied to Lifeact-mEGFP-expressing pollen.

For in vitro pollen germination assays, pollen grains were cultured side-by-side on PGM. Col-0 pollen was cultured on PGM supplemented

with oryzalin (1, 1.5, or 2.5 µM) or BTB-1 (25, 35, or 45 µM) for 1, 2, or 3 h, respectively. Oryzalin (Dr. Ehrenstorfer GmbH, 19044-88-3) or BTB-1 (Selleck, 86030-08-2) was dissolved in DMSO and diluted with PGM. The same dilution ratio of DMSO was added as a control. To quantify the velocity of cytoplasmic streaming and pollen tube growth, Col-0 pollen was placed on PGM containing different concentrations of oryzalin or BTB-1 for 2 h. Due to severe defects of myosin mutant pollen in vesicle trafficking. Col-0, *myo11c1*, *myo11c2*, and *myo11c1 myo11c2* pollens were cultured to reach a length of around 80 µm before images were collected. For VN movement assays, Pro-LAT52:H2B-GFP/ProDUO1:tdTomato pollen was smeared onto solid PGM containing 1.5 µM oryzalin, 15 µM nocodazole (Sigma-Aldrich, 31430-18-9) or 35 µM BTB-1 for 2 h. The *kin14g*, *kin14h* and *kin14g kin14h* double mutants were crossed with the ProLAT52:H2B-GFP/ ProDUO1:tdTomato marker line, respectively. The homozygous pollen was cultured on PGM for 2 h before imaging. To test whether Myo11C1 and Myo11C2 contributed to VN movement, the pollen grains of single mutants of *myo11c1*, *myo11c2* and the double mutants of *myo11c1 myo11c2* were germinated on PGM supplemented with SYBR Green I (1:2000 diluted, Solarbio, SY1020), which can label DNA in VN and the sperm nuclei (SN)[8].

To observe the recovery of cytoskeleton polymerization in pollen tubes after oryzalin treatment, we cultured pollen tubes on a dialysis membrane. The dialysis membrane with pollen was placed in a culture medium for 2 h and transferred to a medium containing 1.5 µM oryzalin for 15 min. For recovery, pollen on the dialysis membrane was transferred to an inhibitor-free liquid medium for several times before the observation was initiated.

To observe the recovery of microtubule polymerization in pollen tubes after oryzalin and BTB-1 co-treatment, the dialysis membrane with pollen was placed in a culture medium with 35 µM BTB-1 for 2 h and transferred to a medium containing 1.5 µM oryzalin and 35 µM BTB-1 for 15 min. For recovery, pollen on the dialysis membrane was transferred to a liquid medium supplemented with 35 µM BTB-1 before the observation was initiated.

For LatB and PBP treatment, the pollen was cultured on PGM for 2 h before the application of 100 nM LatB (Abcam, 76343-94-7) or 625 nM PBP (Adipogen, 10245-81-5). To observe the recovery of actin filaments polymerization in pollen tubes after LatB or LatB and ory-zalin co-treatment, respectively. The dialysis membrane with pollen was placed in a culture medium for 2 h and transferred to a medium containing 100 nM LatB or 100 nM LatB and 1.5 µM oryzalin for 15 min. For recovery, pollen on the dialysis membrane was transferred to inhibitor-free or supplemented with 1.5 µM oryzalin liquid medium for several times before the observation was initiated.

For the recovery of microtubules polymerization in pollen tubes after LatB and oryzalin co-treatment, the dialysis membrane with pollen was placed in a culture medium for 2 h and transferred to a medium containing 100 nM LatB and 1.5 µM oryzalin for 15 min. Then pollen on the dialysis membrane was transferred to the culture medium added with LatB alone.

For labeling mitochondria, the pollen was cultured on PGM for 2 h before the application of 200 nM MitoTracker (Invitrogen, M7512) for 5 min.

## Immunofluorescence assays
Pollen grains and tubes grown on culture medium were incubated in freshly prepared fixing solution containing 3.7% paraformaldehyde, 10% sucrose, 50 mM PIPES, 2 mM EGTA, and 2 mM MgCl$_2$ at pH 6.9 for 30 min at room temperature. Samples were rinsed three times with PEM buffer (50 mM PIPES, 2 mM EGTA, and 2 mM MgCl$_2$, pH 6.9) and treated for 30 min with 2% w/v Driselase (Sigma-Aldrich, 85186-71-6) to digest the cell wall. After rinsing in phosphate-buffered saline (PBS) three times, they were treated with ice-cold methanol at −20 °C for 5 min and rinsed three times with PBS. Nonspecific sites were blocked

by incubation with 1% BSA in PBS at room temperature for 1 h. Pollen grains and tubes were then incubated overnight with the anti-β tubulin primary polyclonal antibody (Abcam, ab15568) diluted at 1:500 in PBS at 4 °C. After three times of rinses with PBS, the samples were incubated for 1 h with goat anti-rabbit IgG Alexa Fluor 594 (Abcam, ab150080) diluted at 1:500 in PBS at 37 °C in the dark. The samples were rinsed with PBS. Optical sections (0.5 µm) and three-dimensional projections of specimens were obtained by an LSM 880 microscope (Zeiss) with a ×63 oil immersion objective.

## Time-lapse imaging and image analysis
Pollen germination rate and pollen tube growth rate were quantified by observing pollen grains and pollen tubes under an IX83 (Olympus, Japan) with a 10×objective. For time-lapse imaging of microtubules, actin filaments, and MGU dynamics during pollen grain germination and tube growth, images were acquired using a high-speed laser confocal living cell workstation (Dragonfly, Andor, Oxford Instruments, UK) equipped with a ×60 1.42 NA oil immersion objective and a Zyla 4.2 Megapixel sCMOS camera (Andor Technology). The 488/561 nm lasers and brightfield illumination were used for imaging. Time-lapse images were collected at 2, 30, or 45 s time intervals with 0.5 µm step intervals, respectively. VN dynamics in *myo11c1*, *myo11c2*, or *kin14g*, *kin14h* single and double mutants were imaged using a spinning disk confocal microscope (UltraView VoX, PerkinElmer, UK) equipped with a Nikon TiE inverted microscope, a Yokogawa Nipkow CSU-X1 spinning disk scanner, ×10 objective and Hamamatsu EMCCD 9100-13. The bright field and 488 nm lasers were used for imaging. Time-lapse images were collected at 30 s time intervals with 0.5 µm intervals. Single-layer imaging of cytoplasmic streaming in wild-type pollen tubes treated with oryzalin, BTB-1, or PBP was observed with an LSM 710 microscope (Zeiss) with a ×63/1.4 oil objective; brightfield illumination was used. Time-lapse images were collected at 1 s time interval. Single-layer imaging of cytoplasmic streaming in *myo11c1 myo11c2* pollen tubes was observed as above described. Indirect immunofluorescence imaging was performed with an LSM 880 microscope (Zeiss) equipped with a ×63/1.4 oil objective, and 405/488/561 nm lasers were used. The acquired images were processed using Imaris (Bitplane), Zen Blue software (Zeiss), and ImageJ (https://imagej.nih.gov/ij/). The relative density was the ratio of the density of the selected area to that of the whole cell[19]. Kymograph analysis was performed using the Kymo-graphBuilder plugin in ImageJ. Trajectory analysis of VN movement was generated with the TrackMate plugin in ImageJ. The Segmented Line plugin in ImageJ was used to measure the velocity of VN. The PIVlab, an open research software for Matlab, was used to analyze and obtain the vector distribution of the velocity of vesicles[47]. The Fibril-Tool plugin in ImageJ was used to quantify the anisotropy of vesicle trafficking[48]. The center of VN was the middle of the black dotted squares. The length of the black dotted squares was 20 µm.

## Statistical and reproducibility
The data represent the mean ± SD based on at least three repeated experiments. The two-sided Student's *t*-test and one-way ANOVA were used to analyze significant differences between groups.

## Reporting summary
Further information on research design is available in the Nature Portfolio Reporting Summary linked to this article.

## Data availability
Biological materials can be obtained upon request. Source data are provided with this paper.

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

## Acknowledgements

We thank Prof. Lijia Qu (Peking University) for generously providing the ProLAT52:H2B-GFP/ProDUO1:tdTomato transgenic line. We acknowledge Prof. Andreas Nebenführ (University of Tennessee) for sharing the *myo11c1* and *myo11c2* double mutants. This work was supported by the National Natural Science Foundation of China (91854206 and 32170335 to H.R.; 32270350 and 32070194 to Y.Z.; 32001522 to F.Z.; 32100297 to T.W.).

## Author contributions

H.R., Y.Z., and X.W. conceived the project and designed the experiments. X.W., T.L., J.X., and L.L. performed experiments. C.W. generated the *Lat52:mCherry-MBD and Lat52:Lifeact-mEGFP lines in wit1 wit2* double mutant background. X.W., H.R., Y.Z., F.Z., T.W., T.L., J.X., and L.L. analyzed the data. X.W., F.Z., Y.Z., and H.R. wrote the manuscript with input from all authors.

## Competing interests

The authors declare no competing interests.
