## [Peer Review File · Nature Communications]

REVIEWER COMMENTS

Reviewer #1 (Remarks to the Author):

In this manuscript, the authors describe that 1) the establishment of the marker line that visualizes microtubules in the Arabidopsis pollen grain and growing pollen tube, 2) overall dynamics of actin filaments and microtubules during pollen germination and tube growth, and 3) the investigation of cytoskeleton and motor protein (myosins and kinesins) functions through drug applications. Based on the results, the authors speculated that cytoplasmic streaming, regulated by actin filaments and myosins, drives the overall movement of MGU, while microtubules and kinesins fine-tune the positioning and directional migration of MGU.

Although the presented story line is clear, the majority of the conclusions is based solely on pharmacological studies. I understand that gene redundancy is one of the major problems in plants for genetic investigation; however, the amount of the data presented in the manuscript, without any genetic characterization, is unfortunately not enough to be convincing. Because the story is clear and interesting, I suggest to 1) check the dynamics of at least one of pollen-expressed myosins and kinesins and investigate whether their dynamics is consistent with the speculated functions and 2) investigate cytoskeleton dynamics in the known mutants which show defects in MGU itself and/or MGU movement like the *wit1;wit2* double mutant (the position of VN and SC can also be altered), *SC-cal* (VN and SC can be separated), and/or the *drop1;drop2* double mutant (no SCs are generated). These genetic experiments would further support pharmacological data if the speculated function of cytoskeletons and motors align with the MGU phenotypes.

References:

<https://doi.org/10.1073/pnas.1323104111>

<https://www.nature.com/articles/s41467-021-22661-8>

<https://www.nature.com/articles/nplants201779>

Minor points:

Line 125: Is this really a ring or sphere? I assume the images are z-projections of a few z-planes and the entire pollen grain image is necessary to check this.

Lines 150-152: sample numbers?

Lines 192-200: sample numbers?

Figure 6a: I cannot see black arrows. I see them in the movie, but the panel images should be updated.

Figure 6: Cytoplasmic streaming can vary based on the position. Did you check the flow in a 3D manner?

Reviewer #2 (Remarks to the Author):

The manuscript by Wang et al. reported that visualization of microtubules in pollen tubes and possible contribution of regulation of the male germ unit transportation. I consider the live imaging of microtubules in elongating pollen tubes significant. The results derived from the pharmacological experiments are informative. Overall, the data are clearly presented. However, the manuscript is descriptive. It ultimately remains unclear what molecular mechanism is responsible for the fine-tuning of VN movement by microtubules and kinesins. I believe more mechanistic data would strengthen this point. My other major comments are listed below.

1. Lifeact signal at the plasma membranes

As mentioned by the authors in lines 126-127, the Lifeact-mEGFP signal appeared at the plasma membranes. Is this signal truly derived from actin filaments? If confocal images of the cell cortex were obtained, would we be able to see a filament organization? If the cells were treated with an actin polymerization inhibitor, such as LatB, would this signal near the plasma membrane disappear? Moreover, if this signal is confirmed to be actin filaments, please include a discussion on the physiological significance of the transient localization.

2. Kymograph analysis

I noticed that the distance to the edge of the pollen grain, as shown in the kymograph in Figure 2e, seems to decrease over time. Is this truly attributable to the shrinking of the pollen grain, or could it be due to drift? If it is the latter, then microscopic image processing techniques such as registration should be applied. Also, wouldn't it be more reader-friendly if the time axis of the kymograph were oriented from top to bottom? I suggest similar changes be implemented for the other kymographs in this manuscript. In addition, it is difficult to see the microtubule entry into the tube. Please improve the presentation e.g. displaying a kymograph with two separate colors.

3. Cytoskeleton localizations in pollen tube apex

The authors have stated, "short microtubules were detected at the position of the actin fringe in the subapical region." However, it did not appear to me that the microtubules and actin filaments were co-localized in the sub-apical region. I would like to see either a rewording or a quantitative

demonstration of co-localization to more accurately depict the localization of the microtubules and actin filaments. In addition, the authors have stated, "At the apical dome, almost no microtubules or actin filaments were observed in these pollen tube." I do not agree with this statement. Especially for actin filaments, it appears to be localized to a certain extent. I guess a quantitative description is needed.

4. Colocalization with the VN leading edge and cytoskeletons

The authors showed the colocalization of the VN leading edge and microtubules as shown in Figure 3e. I found this observation very intriguing. However, I also have some concerns regarding the localization of actin filaments during this process. Is it possible that the VN leading edge is a result of interaction with actin filaments running parallel to the microtubules? I believe it is crucial to detail the differences between microtubules and actin filaments in this manuscript. The authors themselves observed the co-localization of actin filaments and VNs in Supplementary Figure 4. If possible, a triple-staining experiment of VNs, actin fibers, and microtubules would greatly enhance this manuscript. In addition, the line used to make the intensity profiles (Fig. 3f, Supplementary Fig. 4) should be shown.

5. Side effects of the inhibitor treatment

While the phenotype observed under oryzalin treatment is clear and intriguing, it is worth noting that oryzalin has been reported to significantly alter the endoplasmic reticulum and other intracellular membrane systems, in addition to microtubules (Langhans et al., 2009 Protoplasma). There is a concern about the potential for such side effects in this pollen tube experimental system, which may in turn influence the movement of VNs. I recommend that the authors consider using other microtubule inhibitors. Furthermore, employing endomembrane markers could be useful for investigating any side effects caused by oryzalin.

With regard to BTB-1, the effects on germination rate and pollen tube elongation should be checked as with oryzalin. The effects on cytoplasmic streaming (vesicle trafficking) and on actin filament organizations should also be examined for both oryzalin and BTB-1.

6. The VN morphology under the BTB-1 treatment

I found it challenging to fully interpret the results presented in Figure 5g. I did not observe any abnormal nuclear morphology under BTB-1 treatment as illustrated in Figure 5b. Why did the nuclei only elongate in the experimental system that had been pretreated with oryzalin? It is difficult for me to envisage the mechanism by which kinesin is involved in nuclear morphology. The experimental system reminded me of an abnormality in the endoplasmic reticulum and nuclear envelop caused by oryzalin, but additional mechanistic results and discussion are needed to provide clarity on this point.

Reviewer #3 (Remarks to the Author):

The delivery of the male germ unit (MGU) and growth of pollen tubes is of critical importance in angiosperms. This article investigates the relative contributions of the microtubule and actin cytoskeletal networks to this process, using cultured pollen, and live cell imaging, and pharmacological perturbations. The results provide evidence that microtubules have a role in coordinating the motility of the two sperm cells and vegetative nucleus from pollen germination and during pollen tube growth. At this point the results provide a description of how the MGU is trafficked from the onset of germination through extension of the pollen tube. Using specific drugs to depolymerize microtubules and actin filaments, and to inhibit kinesin and myosin motors, the authors suggest that the microtubules that encage the MGU constrain its movement within the vigorous actomyosin-mediated cytoplasmic streaming that is essential for rapid pollen tube growth. Previous work has largely relied on immunofluorescence techniques so the addition of live imaging provides new insight.

The manuscript is well organized, and the experiments conducted appropriately, described in detail in the methods, and with necessary controls, and the conclusions make sense in light of the experimental strategies used.

One potential criticism of the methodology is that this work has relied entirely on two heterologous reporter constructs, a mCherry-labeled microtubule binding domain (mCherry-MBD) and Lifeact-mEGFP, which label but are not endogenous components of microtubules and actin filaments respectively. Thus, it cannot be certain that the organization and specific behaviour of the microtubules and actin filaments, and the MGU, especially in response to drug perturbations, is accurate. Regarding the mCherry-MBD reporter, the authors do assess pollen germination and conclude that the mCherry-MBD does not perturb this process. I think it would be good to also determine if pollen tube growth, can be affected by use of this reporter. It is quite surprising, and unfortunate that the various tubulin reporters developed by the authors did not work satisfactorily, especially when the authors took the necessary steps of driving expression with endogenous promoters, and expressing the reporters in mutant backgrounds to avoid overexpression artefacts. One possible explanation is that the transgene insertion sites are limiting expression levels, so I would encourage screening a larger number of transformants to identify optimal lines (but see next comment; it might be that expression is affected when pollen tubes are growing in vitro.

A second issue is that the assays (necessarily) are taking place in vitro on media-supplemented plates and not in situ, where the pollen tube is having to force its way through stylar tissues. Do these reporter lines affect fertility? Perhaps fertilization rates can be compared to wild-type pollen.

A third concern is the use of drugs. While oryzalin and latrunculin are pretty reliable, the BTB and PBP are likely to be non-specific, targeting different motor proteins. Some cautious interpretation is warranted. For example, while the kinesin inhibitor provided some interesting observations suggesting that kinesin motors could direct motility and even vegetative nucleus morphology, it is not clear how this could be achieved mechanistically, and by which (of many) kinesins. I am not suggesting for the current work that the authors take a genetic approach at this point but recommend some acknowledgement of the deficiencies in mechanistic insight.

One thing I think that should be noted is that microtubule assembly in the pollen grain initiated around the MGU, reminiscent of microtubule nucleation events in recently divided vegetative cells. In contrast, in established pollen tubes, after oryzalin treatment, nucleation began near the pollen tube apex, far away from the MGU. Does this suggest that the MGU loses its capacity to form microtubule arrays once germination is complete?

some minor points:

line 334: "be designed to create" sounds like divine intervention. Perhaps substitute "microtubules may generate enough space".

line 388: replace "vernalized" with "stratified"

line 401: unless you are referring to pollen from different types of plants, "pollen" and not "pollens" is plural.

We appreciate the constructive and insightful suggestions and comments from the three reviewers. We have carefully performed the suggested experiments and revised the manuscript as recommended. All changes have been highlighted in RED font color in the revised manuscript. Please find a detailed point-by-point response below.

Point-by-point response to the reviewers' comments

REVIEWER COMMENTS

Reviewer #1 (Remarks to the Author):

In this manuscript, the authors describe that 1) the establishment of the marker line that visualizes microtubules in the Arabidopsis pollen grain and growing pollen tube, 2) overall dynamics of actin filaments and microtubules during pollen germination and tube growth, and 3) the investigation of cytoskeleton and motor protein (myosins and kinesins) functions through drug applications. Based on the results, the authors speculated that cytoplasmic streaming, regulated by actin filaments and myosins, drives the overall movement of MGU, while microtubules and kinesins fine-tune the positioning and directional migration of MGU.

Although the presented story line is clear, the majority of the conclusions is based solely on pharmacological studies. I understand that gene redundancy is one of the major problems in plants for genetic investigation; however, the amount of the data presented in the manuscript, without any genetic characterization, is unfortunately not enough to be convincing. Because the story is clear and interesting, I suggest to

1) check the dynamics of at least one of pollen-expressed myosins and kinesins and investigate whether their dynamics is consistent with the speculated functions.

R: We fully agree with the Reviewer that additional genetic data would strengthen the conclusions drawn from pharmacological studies. We therefore give priority to analyzing pollen-expressed *kinesin* and *myosin* mutants than checking the dynamics of kinesin and myosin proteins.

KIN14G (At5g27000) and *KIN14H* (At1g09170) are two kinesins in the same subclade of the phylogenetic tree (Richardson DN *et al.*, 2006) and are preferentially expressed in mature pollen (Supplemental Fig. 6a, b). To examine their potential function in MGU migration, we obtained T-DNA insertional mutants of *kin14g* and *kin14h*, respectively. Given that *KIN14G* and *KIN14H* might work redundantly, we crossed *kin14g* and *kin14h* to obtain the *kin14g kin14h* double mutant. The velocity of VN forward movement was significantly increased in *kin14g* single mutants and this phenotype was further enhanced in the *kin14g kin14h* double mutants compared to the control (Fig. 5g). The behavior of VN in the double mutants showed a significantly increased amplitude of both forward and backward movement, leading to an irregular movement pattern (Supplemental Fig. 6c-f). The distance of VN to the pollen tip was also markedly increased in *kin14g kin14h* pollen tubes compared to the control (Fig.5h). These observations were consistent with those observed in the pollen tubes treated with kinesin inhibitors (Fig. 5a-f).

Similar to *KIN14G* and *KIN14H*, *Myo11C1* (At1g08730) and *Myo11C2* (At1g54560) are also highly expressed in mature pollen (Supplemental Fig. 6g, h). The velocity of cytoplasmic streaming and the VN were significantly slowed down in *myo11c1*, *myo11c2* and *myo11c1 myo11c2* pollen tubes compared to the control (Fig. 6l, m). The VN was significantly closer to tip in the single and double mutants than that in the control pollen tubes (Fig. 6n). These observations mimicked those observed in the pollen tubes treated with myosin inhibitors (Fig. 6h- k).

Thus, we have provided genetic data, which are in general consistent with the data obtained from the pharmacological studies. We are in the progress of evaluating the subcellular localization and dynamics of the above-mentioned kinesin and myosin proteins in pollen tubes. Due to the time constraints, we did not manage to include these data into the revised manuscript. However, if the Editor and Reviewer insist that we include these results, we are of course open for such suggestions.

- 2) investigate cytoskeleton dynamics in the known mutants which show defects in MGU itself and/or MGU movement like the *wit1;wit2* double mutant (the position of VN and SC can also be altered), *SC-cal* (VN and SC can be separated), and/or the *drop1;drop2* double mutant (no SCs are generated). These genetic experiments would further support pharmacological data if the speculated function of cytoskeletons and motors align with the MGU phenotypes.

References:

<https://doi.org/10.1073/pnas.1323104111>

<https://www.nature.com/articles/s41467-021-22661-8>

<https://www.nature.com/articles/nplants201779>

R: We thank the Reviewer for the suggestion and have introgressed the microtubule marker, mCherry-MBD, and the actin filament marker, Lifeact-mEGFP, into the *wit1 wit2* double mutants, respectively, as we have this mutant at hands.

To investigate cytoskeleton dynamics, we collected time-lapse images of mCherry-MBD and Lifeact-mEGFP, respectively. The global dynamics were examined through a correlation coefficient analysis (Vidali L *et al.*, 2010). This analysis quantified the extent of overall cytoskeleton rearrangements over time by calculating the correlation of fluorescent intensity at all pixel positions between all pairwise temporal intervals. The change of cytoskeleton organization over time was reflected by the rate of decay of correlation coefficient values as the temporal interval increased. The correlation coefficient curves decayed similarly in *wit1 wit2* double mutants as in the control pollen tubes (Supplemental Fig. 5a-d). These data indicated that neither actin filament dynamics or microtubule dynamics were attenuated in *wit1 wit2* pollen tubes. It has been reported that WIT proteins can physically interact with myosin proteins (K Tamura *et al.*, 2013). We thus speculate that the VN transport defect in *wit1 wit2* double mutants is likely due to impaired connection of VN with cytoskeleton rather than cytoskeleton dynamics. This point has been discussed in the revised manuscript (Lines 417- 420).

Minor points:

Line 125: Is this really a ring or sphere? I assume the images are z-projections of a few z-planes and the entire pollen grain image is necessary to check this.

R: The images in Fig. 3a, b are single frame images, but not z-projections. It is ring shape in these images. As suggested, sequential z-plane images of the entire pollen grain have been collected. As shown below, microtubules were indeed assembled around ring (sphere in 3D)- like structures (the VNs labelled by H2B-GFP) in pollen grains shortly after hydration.

Lines 150-152: sample numbers?

R: The sample number is 30 pollen tubes, which has been clarified in the revised manuscript (Line 162).

Lines 192-200: sample numbers?

R: The sample number is 5 pollen tubes, which has been clarified in the revised manuscript (Line 216).

Figure 6a: I cannot see black arrows. I see them in the movie, but the panel images should be updated.

R: We thank the Reviewer for pointing this out. As suggested, the panel images have been updated in the revised manuscript (Fig. 6a).

Figure 6: Cytoplasmic streaming can vary based on the position. Did you check the flow in a 3D manner?

R: We agree with the Reviewer that the cytoplasmic streaming can vary in different regions of pollen tubes. We thus examined the cytoplasmic streaming around the VN. Single-frame imaging was applied for this analysis, because the cytoplasmic streaming is too fast to track the particles in a 3D manner.

Reviewer #2 (Remarks to the Author):

The manuscript by Wang et al. reported that visualization of microtubules in pollen tubes and possible contribution of regulation of the male germ unit transportation. I consider the live imaging of microtubules in elongating pollen tubes significant. The results derived from the pharmacological experiments are informative. Overall, the data are clearly presented. However, the manuscript is descriptive. It ultimately remains unclear what molecular mechanism is responsible for the fine-tuning of VN movement by microtubules and kinesins. I believe more mechanistic data would strengthen this point.

R: We thank the Reviewer for the insightful comments and suggestions. The lack of genetic data is concerned by all three reviewers. We therefore have performed additional experiments with kinesin mutants and myosin mutants. As shown in Fig. 5g, h; Supplemental Fig. 6c-f, an irregular movement pattern of VN was observed in *kin14g kin14h* pollen tubes compared to the control. Although these data have not elucidated the mechanism, they at least supported that the two kinesin proteins, KIN14G and KIN14H, were involved in fine-tuning of VN movement in pollen tubes. Due to the time constraints, we hope that the Editor and Reviewer agree that detailed analysis of these kinesin and myosin proteins can be studied as an excellent future project. We have therefore opted not to include further analysis on these proteins in the current manuscript. Instead, we aim to focus more on characterizing the distinct functions of microtubules and actin filaments and how they coordinate in the transportation of the male germ units in this paper.

My other major comments are listed below.

1. Lifeact signal at the plasma membranes

As mentioned by the authors in lines 126-127, the Lifeact-mEGFP signal appeared at the plasma membranes. Is this signal truly derived from actin filaments?

R: Yes, the signal is derived from actin filaments based on the LatB-treatment experiment. The Lifeact-mEGFP signal near the plasma membrane is markedly reduced after LatB treatment compared to the control (Fig. 2c), supporting that this signal is derived from actin filaments.

If confocal images of the cell cortex were obtained, would we be able to see a filament organization?

R: Yes, we could observe evident actin filament organization at cell cortex (Fig. 2c).

If the cells were treated with an actin polymerization inhibitor, such as LatB, would this signal near the plasma membrane disappear?

R: Yes, the Lifeact-mEGFP signal near the plasma membrane is markedly reduced after LatB treatment compared to the control (Fig. 2c).

Moreover, if this signal is confirmed to be actin filaments, please include a discussion on the physiological significance of the transient localization.

R: We thank the Reviewer for the nice suggestion. We have added discussion on this point in the revised manuscript (Lines 382- 383).

2. Kymograph analysis

I noticed that the distance to the edge of the pollen grain, as shown in the kymograph in Figure 2e, seems to decrease over time. Is this truly attributable to the shrinking of the pollen grain, or could it be due to drift? If it is the latter, then microscopic image processing techniques such as registration should be applied.

R: The Reviewer is correct that the plasma membrane of the germination site will first invaginated before it protruded. We have described this phenomenon in our recent paper (Ruan HQ *et al.*, 2023). To further ruling out the possibility of drift, we performed kymograph analysis at another edge of the pollen grain, as labeled by the white line in the images below (Fig. a). There was no change of position for this edge, suggesting no drift of the pollen (Fig. b).

Also, wouldn't it be more reader-friendly if the time axis of the kymograph were oriented from top to bottom? I suggest similar changes be implemented for the other kymographs in this manuscript.

R: We thank the Reviewer for the nice suggestion. We have modified all the kymographs in the manuscript as suggested, including Fig. 2f; Fig. 4f, g, I; Fig. 5c, d, j; Fig. 6i; Supplemental Fig. 4a, b; Supplemental Fig. 6c-f, i-l; Supplemental Fig. 8b, f, g.

In addition, it is difficult to see the microtubule entry into the tube. Please improve the presentation e.g. displaying a kymograph with two separate colors.

R: The kymograph has been revised as suggested (Fig. 2f). The Reviewer is correct that at this time point, actin filaments have entered the tube, but microtubules might have not yet entered. With the time, the microtubules will eventually enter the pollen tube.

3. Cytoskeleton localizations in pollen tube apex

The authors have stated, "short microtubules were detected at the position of the actin fringe in the subapical region." However, it did not appear to me that the microtubules and actin filaments were co-localized in the sub-apical region. I would like to see either a rewording or a quantitative demonstration of co-localization to more accurately depict the localization of the microtubules and actin filaments.

R: We thank the Reviewer for pointing this out. We have analyzed the colocalization of mCherry-MBD and mEGFP-Lifeact in the sub-apical region of the dual-labeled pollen tubes. As shown in Fig. 2h and 2i, mCherry-MBD and mEGFP-Lifeact did not co-localized well at the subapical region. We thus reworded the description to be “short microtubules were detected in the subapical region, where actin filaments formed the actin fringe” in the revised manuscript (Lines 147- 148).

In addition, the authors have stated, "At the apical dome, almost no microtubules or actin filaments were observed in these pollen tube." I do not agree with this statement. Especially for actin filaments, it appears to be localized to a certain extent. I guess a quantitative description is needed.

R: We thank the Reviewer for pointing this out. The Reviewer is correct that a few actin filaments indeed exist in the apical region of pollen tubes. The description has been revised to be “At the apical dome, while a few actin filaments existed, microtubules were hardly observed in these pollen tubes” in the revised manuscript (Lines 148- 149; Fig. 2j, k).

4. Colocalization with the VN leading edge and cytoskeletons

The authors showed the colocalization of the VN leading edge and microtubules as shown in Figure 3e. I found this observation very intriguing. However, I also have some concerns regarding the localization of actin filaments during this process. Is it possible that the VN leading edge is a result of interaction with actin filaments running parallel to the microtubules? I believe it is crucial to detail the differences between microtubules and actin filaments in this manuscript. The authors themselves observed the co-localization of actin filaments and VNs in Supplementary Figure 4. If possible, a triple-staining experiment of VNs, actin fibers, and microtubules would greatly enhance this manuscript.

R: We thank the Reviewer for the insightful comments and suggestions. To visualize the VN, actin filaments and microtubules at the same time in living cells, we tried our best to stain the VN with Hoechst 33342 in the dual fluorescent pollen tubes of mCherry-MBD and Lifeact-mEGFP. Unfortunately, Hoechst could not entry the pollen tubes, even at high concentration and long incubation time.

We agree with the Reviewer that it is possible that the VN leading edge is a result of interaction with actin filaments running parallel to the microtubules. We have added discussion on this point in the revised manuscript (Lines 376- 379).

In addition, the line used to make the intensity profiles (Fig. 3f, Supplementary Fig. 4) should be shown.

R: The lines have been added as suggested (Fig. 3e, Supplemental Fig. 7a).

5. Side effects of the inhibitor treatment

While the phenotype observed under oryzalin treatment is clear and intriguing, it is worth noting that oryzalin has been reported to significantly alter the endoplasmic reticulum and

other intracellular membrane systems, in addition to microtubules (Langhans et al., 2009 Protoplasma). There is a concern about the potential for such side effects in this pollen tube experimental system, which may in turn influence the movement of VNs. I recommend that the authors consider using other microtubule inhibitors.

R: We thank the Reviewer for the comments and suggestions. As suggested, we have applied another widely used microtubule inhibitor, nocodazole (Idilli Al *et al.*, 2013). Application of nocodazole resulted in an increased amplitude of both forward and backward movement of the VN, leading to an irregular movement pattern (Supplemental Fig. 4a, b). Both forward and backward movement were significantly faster in the presence of nocodazole than in the control (Supplemental Fig. 4c). These observations were in consistent with the data obtained from oryzalin-treated pollen tubes (Fig. 4d- h).

Furthermore, employing endomembrane markers could be useful for investigating any side effects caused by oryzalin.

R: To check whether oryzalin treatment impacted on intracellular membrane systems, we stained mitochondria with the fluorescent dye, MitoTracker, in pollen tubes. We also tried to stain the ER with the dye, ER-Tracker Blue-White DPX. However, ER-Tracker Blue-White DPX. did not work well in pollen tubes. Disassembly of microtubules had minor impact on mitochondria movement, as the velocity of mitochondria movement was not significantly altered in the presence of oryzalin or nocodazole compared to the control (Supplemental Fig. 4e). In addition, we have performed experiments with kinesin mutants. Similar to the observation in pollen tubes treated with oryzalin or BTB-1, an irregular movement pattern of VN was observed in *kin14g kin14h* pollen tubes compared to the control (Supplemental Fig. 6f). Together, these data supported that the effect of oryzalin treatment on VN movement is from its effect on microtubules rather than side effect on intracellular membrane systems.

With regard to BTB-1, the effects on germination rate and pollen tube elongation should be checked as with oryzalin.

R: We have evaluated the effect of BTB-1 on pollen germination rate and pollen tube elongation as suggested. Similar to oryzalin treatment, application of BTB-1 promoted pollen germination, but had minor impact on pollen tube elongation (Supplemental Fig. 3d-f).

The effects on cytoplasmic streaming (vesicle trafficking) and on actin filament organizations should also be examined for both oryzalin and BTB-1.

R: We thank the Reviewer for the suggestion. Quantification analysis suggested that both the cytoplasmic streaming and actin filament organizations was not significantly affected by oryzalin or BTB-1 treatment (Supplemental Fig. 3g-k).

6. The VN morphology under the BTB-1 treatment

I found it challenging to fully interpret the results presented in Figure 5g. I did not observe

any abnormal nuclear morphology under BTB-1 treatment as illustrated in Figure 5b. Why did the nuclei only elongate in the experimental system that had been pretreated with oryzalin?

R: We thank the Reviewer for pointing this out. We have enlarged the images to show the elongated VN after BTB-1 treatment (Fig. 5b, 0 min, 13 min and 16 min).

It is difficult for me to envisage the mechanism by which kinesin is involved in nuclear morphology. The experimental system reminded me of an abnormality in the endoplasmic reticulum and nuclear envelop caused by oryzalin, but additional mechanistic results and discussion are needed to provide clarity on this point.

R: We proposed that inhibition of kinesin activities or depolymerization of microtubules somehow impaired the coordination between the movement of sperm cells and the VN. When the direction or velocity between the sperm cells and the VN were not matched, the sperm cells will drag the VN to deform it. This hypothesis could explain why the abnormal VN morphology were occasionally, but not constantly, observed. We have added the discussion on this point in the revised manuscript (Lines 398- 401).

Reviewer #3 (Remarks to the Author):

The delivery of the male germ unit (MGU) and growth of pollen tubes is of critical importance in angiosperms. This article investigates the relative contributions of the microtubule and actin cytoskeletal networks to this process, using cultured pollen, and live cell imaging, and pharmacological perturbations. The results provide evidence that microtubules have a role in coordinating the motility of the two sperm cells and vegetative nucleus from pollen germination and during pollen tube growth. At this point the results provide a description of how the MGU is trafficked from the onset of germination through extension of the pollen tube. Using specific drugs to depolymerize microtubules and actin filaments, and to inhibit kinesin and myosin motors, the authors suggest that the microtubules that encage the MGU constrain its movement within the vigorous actomyosin-mediated cytoplasmic streaming that is essential for rapid pollen tube growth. Previous work has largely relied on immunofluorescence techniques so the addition of live imaging provides new insight.

The manuscript is well organized, and the experiments conducted appropriately, described in detail in the methods, and with necessary controls, and the conclusions make sense in light of the experimental strategies used.

One potential criticism of the methodology is that this work has relied entirely on two heterologous reporter constructs, a mCherry-labeled microtubule binding domain (mCherry-MBD) and Lifeact-mEGFP, which label but are not endogenous components of microtubules and actin filaments respectively. Thus, it cannot be certain that the organization and specific behaviour of the microtubules and actin filaments, and the MGU, especially in response to drug perturbations, is accurate.

R: We thank the Reviewer for the comments. We agree with the Reviewer that mCherry-MBD and Lifeact-mEGFP might not be the perfect markers for microtubules and actin filaments. However, they are the best ones available, according to the experiences from our lab and other labs in the field. That's why fluorescently labelled MBD and Lifeact

have been widely used to label microtubules and actin filaments, respectively, in plant cells (Qin T *et al.*, 2014; Qu X *et al.*, 2013; Liu C *et al.*, 2018; Liu C *et al.*, 2021; Ali MF *et al.*, 2023; Hamant O *et al.*, 2008; Marc J *et al.*, 1998; Buschmann H *et al.*, 2006; Stöckle D *et al.*, 2022). For Lifeact-mEGFP, it could clearly label the actin fringes in pollen tubes and have minor impact on pollen germination and pollen tube growth (Supplemental Fig. 2a-c; Liu C *et al.*, 2018; Liu C *et al.*, 2021; Qin T *et al.*, 2014; Qu X *et al.*, 2013). For mCherry-MBD, it successfully labels microtubules in both pollen grains and pollen tubes, which are validated by immunofluorescence assay using antibodies against β -tubulin (Fig. 1m).

Regarding the mCherry-MBD reporter, the authors do assess pollen germination and conclude that the mCherry-MBD does not perturb this process. I think it would be good to also determine if pollen tube growth, can be affected by use of this reporter.

R: As suggested, we have evaluated the pollen tube growth rate in the mCherry-MBD and the mEGFP-MBD reporter lines. The pollen growth rate was also determined in the Lifeact-mCherry- and Lifeact-mEGFP-expressing lines. As shown in Supplemental Fig. 2c the pollen tube growth rates were similar in all these lines as the WT pollen tubes.

It is quite surprising, and unfortunate that the various tubulin reporters developed by the authors did not work satisfactorily, especially when the authors took the necessary steps of driving expression with endogenous promoters, and expressing the reporters in mutant backgrounds to avoid overexpression artefacts. One possible explanation is that the transgene insertion sites are limiting expression levels, so I would encourage screening a larger number of transformants to identify optimal lines (but see next comment; it might be that expression is affected when pollen tubes are growing in vitro).

R: We thank the Reviewer for the comments and suggestions. We have screened a lot of transformants and obtained at least 150 positive transformants for each construct. However, none of them showed high levels of fluorescence.

A second issue is that the assays (necessarily) are taking place in vitro on media-supplemented plates and not in situ, where the pollen tube is having to force its way through stylar tissues. Do these reporter lines affect fertility? Perhaps fertilization rates can be compared to wild-type pollen.

R: No, the reporter lines, including mCherry-MBD, mEGFP-MBD, Lifeact-mCherry or Lifeact-mEGFP, do not affect fertility rates (Supplemental Fig. 2d, e).

A third concern is the use of drugs. While oryzalin and latrunculin are pretty reliable, the BTB and PBP are likely to be non-specific, targeting different motor proteins. Some cautious interpretation is warranted. For example, while the kinesin inhibitor provided some interesting observations suggesting that kinesin motors could direct motility and even vegetative nucleus morphology, it is not clear how this could be achieved mechanistically, and by which (of many) kinesins. I am not suggesting for the current work that the authors take a genetic approach at this point but recommend some acknowledgement of the deficiencies in mechanistic insight.

R: Reviewer 1 and 2 also raised similar concern on the pharmacological experiments. We have performed additional experiments with *kinesin* and *myosin* mutants. The irregular patten of VN movement observed in these mutants were generally in line with the observations after pharmacological perturbations. However, we did not observe elongated VN in the *kin14g kin14h* pollen tubes, suggesting that the VN morphology might be regulated by other kinesin proteins. As suggested, we have added acknowledgement of the deficiencies in mechanistic insight as "Detailed analysis of the functions of KIN14G and KIN14H, as well as other kinesins involved, is required to provide mechanistic insight into the complex process of MGU transport in pollen tubes" in the revised manuscript (Lines 422- 429).

One thing I think that should be noted is that microtubule assembly in the pollen grain initiated around the MGU, reminiscent of microtubule nucleation events in recently divided vegetative cells. In contrast, in established pollen tubes, after oryzalin treatment, nucleation began near the pollen tube apex, far away from the MGU. Does this suggest that the MGU loses its capacity to form microtubule arrays once germination is complete?

R: We thank the Reviewer for the insightful comments. The fluorescence of mCherry-MBD and mEGFP-MBD around the MGU was very weak. We do see some weak signal of mCherry-MBD around the MGU during the recovery of microtubules after washing oryzalin (Fig. 5i, 0s, 320s and 560s). However, when the cytosolic microtubules were assembled, the fluorescence of these microtubules was strong, which challenged the observation of microtubule signals around the MGU. We thus could not fully rule out the possibility that some microtubules may form from the MGU during pollen tube elongation.

some minor points:

line 334: "be designed to create" sounds like devine intervention. Perhaps substitute "microtubules may generate enough space".

R: This has been done (Line 391).

line 388: replace "vernalized" with "stratified"

R: This has been done (Line 456).

line 401: unless you are referring to pollen from different types of plants, "pollen" and not "pollens" is plural.

R: This has been corrected (Line 293, 471, 488, 494, 501, 506, 862, 911, 935, 1019).

** See Nature Portfolio's author and referees' website at www.nature.com/authors for information about policies, services and author benefits.

References

1. Richardson DN, Simmons MP, Reddy ASN. Comprehensive comparative analysis of kinesins in photosynthetic eukaryotes. *Bmc Genomics* **7**, (2006).
2. Vidali L, Burkart GM, Augustine RC, Kerdavid E, Tuezal E, Bezanilla M. Myosin XI Is

- Essential for Tip Growth in *Physcomitrella patens*. *Plant Cell* **22**, 1868-1882 (2010).
3. K Tamura *et al.* Myosin XI-i Links the Nuclear Membrane to the Cytoskeleton to Control Nuclear Movement and Shape in Arabidopsis. *Curr Bio* **23**:1776-1781 (2013).
 4. Ruan HQ, Wang T, Ren HY, Zhang Y. AtFH5-labeled secretory vesicles-dependent calcium oscillation drives exocytosis and stepwise bulge during pollen germination. *Cell Rep* **42**:113319 (2023).
 5. Idilli AI, Morandini P, Onelli E, Rodighiero S, Caccianiga M, Moscatelli A. Microtubule depolymerization affects endocytosis and exocytosis in the tip and influences endosome movement in tobacco pollen tubes. *Mol Plant* **6**, 1109-1130 (2013).
 6. Qin T, Liu X, Li J, Sun J, Song L, Mao T. Arabidopsis Microtubule-Destabilizing Protein 25 Functions in Pollen Tube Growth by Severing Actin Filaments. *Plant Cell* **26**, 325-339 (2014).
 7. Qu X, Zhang H, Xie Y, Wang J, Chen N, Huang S. Arabidopsis Villins Promote Actin Turnover at Pollen Tube Tips and Facilitate the Construction of Actin Collars. *Plant Cell* **25**, 1803-1817 (2013).
 8. Liu C, Zhang Y, Ren H. Actin Polymerization Mediated by AtFH5 Directs the Polarity Establishment and Vesicle Trafficking for Pollen Germination in Arabidopsis. *Mol Plant* **11**, 1389-1399 (2018).
 9. Liu C, Zhang Y, Ren H. Profilin promotes formin-mediated actin filament assembly and vesicle transport during polarity formation in pollen. *Plant Cell* **33**, 1252-1267 (2021).
 10. Ali MF *et al.* Cellular dynamics of coenocytic endosperm development in Arabidopsis thaliana. *Nat Plants* **9**, 330-342 (2023).
 11. Hamant O *et al.* Developmental Patterning by Mechanical Signals in Arabidopsis. *Science* **322**, 1650-1655 (2008).
 12. Marc J *et al.* A GFP-MAP4 reporter gene for visualizing cortical microtubule rearrangements in living epidermal cells. *Plant Cell* **10**, 1927-1940 (1998).
 13. Buschmann H, Chan J, Sanchez-Pulido L, Andrade-Navarro MA, Doonan JH, Lloyd CW. Microtubule-associated AIR9 recognizes the cortical division site at preprophase and cell-plate insertion. *Curr Biol* **16**, 1938-1943 (2006).
 14. Stöckle D *et al.* Microtubule-based perception of mechanical conflicts controls plant organ morphogenesis. *Sci Adv* **8**, eabm4974, (2022).

REVIEWER COMMENTS

Reviewer #1 (Remarks to the Author):

The revised manuscript, incorporating genetics, has significantly enhanced the reliability of the work. The authors have appropriately addressed my comments, except for the 'ring' structure terminology. While it may appear as a ring in a single-plane image, the ring does not exist in the pollen grain. The authors should refrain from labeling it as a 'ring' structure simply because it is misleading.

Additional Comments from Reviewer #1:

I briefly checked reviewer #3 comments and the authors' responses.

Overall, the authors have addressed all comments effectively.

Here, I outline my thoughts on each "Major comment" point.

The first one:

As the reviewer noted, it is correct that markers do not necessarily reflect native cellular dynamics. For instance, the lifeact-FP does not visualize filaments effectively. However, as the authors mentioned, the markers they utilized are the most suitable for investigating their dynamics. As long as the observed phenotypes align with the visualized dynamics, I am comfortable with this approach, so I do not consider it a major issue in this manuscript. Additionally, the authors have utilized microtubule immunostaining as backup evidence. Lifeact is a well-characterized marker for pollen tube F-actin dynamics.

The second one:

The authors' response is satisfactory.

The third one:

The authors' response is satisfactory.

The fourth one:

The authors' response is satisfactory.

The fifth one:

The authors' response is satisfactory.

The sixth one:

It seems that the authors did not incorporate this part into a revised manuscript. The authors should add this to the discussion.

Reviewer #2 (Remarks to the Author):

This manuscript has improved significantly. The addition of results from mutants notably enhances the overall value of the paper. I am generally satisfied with this revised version and appreciate the authors' efforts. However, I would like to suggest further revisions on the following points:

1) The results from the kinesin mutants are crucial and should be highlighted more prominently. I recommend incorporating the time-lapse images and kymographs of the kin14g kin14h double mutant (currently in Supplemental Figure 6f) into Figure 5.

2) In Supplemental Figure 5, the authors should present analysis results of a cell in which cytoskeletal movement is markedly inhibited as a negative control in image correlation analysis. The use of myosin or kinesin inhibitors might be beneficial for this purpose.

3) I would like to see a summary of this study presented as an illustrative diagram, to be added as a supplemental figure. Ideally, this illustration should clearly depict the relationships between actomyosin-dependent cytoplasmic streaming, the microtubule-kinesin system, and the vegetative nucleus (VN), providing an immediate visual understanding of these connections.

We appreciate the nice suggestions and comments from the reviewers. We have carefully performed the suggested experiments and revised the manuscript as recommended. All changes have been highlighted in RED font color in the revised manuscript. Please find a detailed point-by-point response below.

Point-by-point response to the reviewers' comments

REVIEWER COMMENTS

Reviewer #1 (Remarks to the Author):

The revised manuscript, incorporating genetics, has significantly enhanced the reliability of the work. The authors have appropriately addressed my comments, except for the 'ring' structure terminology. While it may appear as a ring in a single-plane image, the ring does not exist in the pollen grain. The authors should refrain from labeling it as a 'ring' structure simply because it is misleading.

R: We thank the Reviewer for pointing this out and have changed the labeling to be “sphere-shaped structures” in the revised manuscript (Lines 128, 151).

Additional Comments from Reviewer #1:

I briefly checked reviewer #3 comments and the authors' responses. Overall, the authors have addressed all comments effectively. Here, I outline my thoughts on each “Major comment” point.

The first one: As the reviewer noted, it is correct that markers do not necessarily reflect native cellular dynamics. For instance, the lifeact-GFP does not visualize filaments effectively. However, as the authors mentioned, the markers they utilized are the most suitable for investigating their dynamics. As long as the observed phenotypes align with the visualized dynamics, I am comfortable with this approach, so I do not consider it a major issue in this manuscript. Additionally, the authors have utilized microtubule immunostaining as backup evidence. Lifeact is a well-characterized marker for pollen tube F-actin dynamics.

The second one:

The authors' response is satisfactory.

The third one:

The authors' response is satisfactory.

The fourth one:

The authors' response is satisfactory.

The fifth one:

The authors' response is satisfactory.

The sixth one:

It seems that the authors did not incorporate this part into a revised manuscript. The authors should add this to the discussion.

R: The discussion on this point has been added in the revised manuscript as suggested (Lines 389- 393).

Reviewer #2 (Remarks to the Author):

This manuscript has improved significantly. The addition of results from mutants notably enhances the overall value of the paper. I am generally satisfied with this revised version and appreciate the authors' efforts. However, I would like to suggest further revisions on the following points:

1) The results from the kinesin mutants are crucial and should be highlighted more prominently. I recommend incorporating the time-lapse images and kymographs of the *kin14g kin14h* double mutant (currently in Supplemental Figure 6f) into Figure 5.

R: We thank the Reviewer for the suggestion. The results about *kin14g kin14h* double mutants have been incorporated into Figure 5 in the revised manuscript as suggested (Fig. 5g-j).

2) In Supplemental Figure 5, the authors should present analysis results of a cell in which cytoskeletal movement is markedly inhibited as a negative control in image correlation analysis. The use of myosin or kinesin inhibitors might be beneficial for this purpose.

R: We thank the Reviewer for the suggestion. We have evaluated the effect of myosin inhibitors, PBP, and the kinesin inhibitor, BTB-1, on cytoskeleton dynamics as suggested. The cytoskeleton dynamics were also re-examined with the same microscope settings in the WT and *wit1 wit2* pollen tubes. The correlation coefficient curves decayed similarly in the presence of motor protein inhibitors as in the control pollen tubes (Supplementary Fig. 5e-h). These data indicated that neither microtubule dynamics or actin filament dynamics was attenuated in the inhibitor-treated pollen tubes.

3) I would like to see a summary of this study presented as an illustrative diagram, to be added as a supplemental figure. Ideally, this illustration should clearly depict the relationships between actomyosin-dependent cytoplasmic streaming, the microtubule-kinesin system, and the vegetative nucleus (VN), providing an immediate visual understanding of these connections.

R: We thank the Reviewer for the nice suggestion. An illustrative diagram has been added in the revised manuscript (Supplementary Fig. 9).

Supplementary Figure 9. The proposed working model of the dynamic interplay among

cytoplasmic streaming, the microtubule-kinesin system, and the vegetative nucleus (VN) in pollen tubes.

In the rapidly growing pollen tubes, the actomyosin-dependent cytoplasmic streaming has significant impact on the velocity and direction of local VN movement, whereas the microtubule-kinesin system finely controls the speed and positioning of VN, allowing the VN to follow and keep pace with the growth of pollen tubes (left panel). Depolymerization of microtubules has minor effect on the cytoplasmic streaming or pollen tube growth. However, the VN loses its stable position to the pollen tube tip and moves in an irregular pattern in pollen tubes in the absence of microtubules (middle panel). When actin filaments are depolymerized, the cytoplasmic streaming, pollen tube growth and VN movement are inhibited (right panel). The blue arrows around the VN indicates the direction of the cytoplasmic circulation encountered by VN. The yellow arrows indicate that the net direction of VN movement toward pollen tube tip is controlled by kinesin. The question marks indicate the possibility that myosin might locate at VN surface and directly participate in VN migration in pollen tubes.

REVIEWERS' COMMENTS

Reviewer #2 (Remarks to the Author):

Since all of my concerns have been more than satisfactorily addressed, and the additional data and discussions introduced in response to my comments have significantly enhanced the paper, I am pleased to recommend this manuscript for publication.